# FINE-TUNING OF TRANSFORMER MODELS WITH FRAMES

## ABSTRACT

Fine-tuning large-scale pre-trained models for downstream tasks remains a challenge, particularly as model sizes continue to grow. While Parameter-Efficient Fine-Tuning (PEFT) strategies such as Low-Rank Adaptation (LoRA) have emerged as effective solutions, their memory requirements scale linearly with the size of the model, $\mathcal{O}(dr)$, where $d$ is the hidden dimension of the model and $r$ is the rank. In this work, we present FrameFT, a novel PEFT method based on Fusion Frames. We model the parameter update $\Delta W$ with a sparse coefficient matrix in the Fusion Frame representation space. It turns out that Fusion Frames can be generated algorithmically and shared across model layers, enabling highly efficient updates. Hence, only the sparse coefficients of the basis expansion are stored/optimized, dramatically reducing the memory footprint and parameter count. The sparse structure of the coefficient matrix in FrameFT, together with the sparsity in the Fusion Frames themselves, provides computational benefits compared to other fine-tuning methods. Our technical analysis shows that FrameFT allows obtaining formal convergence results. We evaluate our method across a suite of supervised fine-tuning benchmarks, primarily focusing on Language tasks, but also report applicability to Vision models. Our empirical evaluation demonstrates that FrameFT achieves performance on par with or exceeding that of state-of-the-art PEFT techniques, while requiring much fewer trainable parameters and memory.

## 1    INTRODUCTION

Foundation models show impressive capabilities across a range of domains, including language (Touvron et al., 2023; Team et al., 2024a; OpenAI, 2025), vision (Dehghani et al., 2023), and other modalities (Łajszczak et al., 2024; Fang et al., 2025). While increasing model size yields improved performance, many downstream applications benefit from an additional fine-tuning step to specialize the model for specific tasks. However, fine-tuning all parameters of a large model is computationally expensive and memory-intensive. This becomes worse when fine-tuning is required for multiple tasks, each potentially needing to store its own dedicated copy of the model.

To address these limitations, Parameter-Efficient Fine-Tuning (PEFT) methods have emerged as a popular alternative (Hu et al., 2022; Zhang et al., 2025; Dettmers et al., 2023). These techniques seek to minimize the number of trainable parameters during fine-tuning, significantly reducing resource needs. Recent research has shown that PEFT methods can match the performance of full-model fine-tuning while updating only a small subset of the model's parameters.

**LoRA and Sparse Fine-tuning.**    The most widely used PEFT technique is Low-Rank Adaptation (LoRA) (Hu et al., 2022), which keeps the pre-trained weights frozen and injects task-specific low-rank matrices into each layer. These matrices efficiently capture necessary adaptations while maintaining low memory and compute costs. LoRA's success has inspired a range of variants that offer improvements in convergence, efficiency, and storage overhead (Dettmers et al., 2023; Hayou et al., 2024; Zhu et al., 2024; Yaras et al., 2024; Xia et al., 2024).

In contrast, sparse fine-tuning (SFT) methods adopt a different strategy: they selectively update a sparse subset of the model's original parameters. The selection criteria vary – some approaches rely on magnitude-based pruning (Lu et al., 2024), while others use sensitivity measures such as Fisher

information (Guo et al., 2021), Fourier-domain analyses (Gao et al., 2024), or parameter change heuristics (Ansell et al., 2022). SFT techniques can sometimes face scalability issues when applied to large models, especially in identifying sparse patterns and tuning hyperparameters (Guo et al., 2021). Some methods also rely on access to training dynamics (Ansell et al., 2022) that may not be easily available.

**Motivation.** LoRA parameterizes the weight updates in a low-dimension subspace spanned by the top few singular vectors. On the other hand, SFT methods constrain the weight updates by selecting a few non-zero coefficients. So, a natural question is: can we design an effective parameterization that **(i)** captures the low-dimensional subspace nature of LoRA, while **(ii)** also keeping the parameter count low by utilizing structured sparsity and **(iii)** offer compute benefits from *how* the weight updates are factorized? Since we are dealing with different subspaces and studying parameterizations involving those subspaces, a particularly relevant concept is the theory of *Finite Frames*. Briefly, Finite Frames and Fusion Frames are used to study the spanning sets and spanning subspaces of a given finite-dimensional vector space, respectively. They have numerous applications in coding theory, compressed sensing (Boufounos et al., 2009), quantization (Adepu et al., 2024; Czaja & Na, 2024), and dictionary learning (Hwang et al., 2019; Cai et al., 2014). Their ability to encode updates in overlapping subspaces offers robustness, flexibility, and efficiency – properties that are valuable for model adaptation. Our FrameFT will leverage these properties by partitioning the model's parameter space into fusion-frame-based subspaces, within which updates are learned in a sparse and structured manner.

**Our proposal: Fine-tuning with Frames.** We propose a new PEFT strategy inspired by Fusion Frames (Christensen, 2018; Casazza et al., 2008; Waldron, 2019). Fusion Frames allow structured representations by decomposing a space into overlapping subspaces at multiple scales. Unlike strategies that rely on either low-rank or sparse updates, FrameFT will perform fine-tuning across multiple structured subspaces, naturally combining the global adaptability of LoRA with the precision of sparse updates.

**Contributions.** Our main contributions are: **(a)** A new fine-tuning framework, **FrameFT**, that utilizes highly structured subspaces (Fusion Frames) to capture parameter updates efficiently; **(b)** Extensive empirical evaluations across multiple foundation models and tasks, demonstrating FrameFT's effectiveness compared to state-of-the-art PEFT baselines; **(c)** A theoretical analysis of the numerical stability, convergence and efficiency benefits offered by FrameFT.

## 2 SUBSPACE DECOMPOSITIONS AND FRAME THEORY

Here, we provide a brief review of Finite Frame theory for Hilbert spaces. A reader familiar with these concepts can skip to the next section, where we describe applying this idea to fine-tuning.

**Spanning sets and Frames.** A spanning set is a collection of vectors that spans a finite-dimensional Hilbert space or vector space, and orthonormal bases are a canonical example. Frames generalize this concept by allowing redundancy, enabling stable and often more robust representations.

Let $\mathcal{H}^d$ denote a $d$-dimensional Hilbert space. A sequence of vectors $\phi = (\varphi_i)_{i=1}^k$ in $\mathcal{H}^d$ is called a frame if there exist constants $0 < A \leq B < \infty$ such that for all $x \in \mathcal{H}^d$,

$$A||x||^2 \leq \sum_{i=1}^k |\langle x, \varphi_i \rangle|^2 \leq B||x||^2, \quad (1)$$

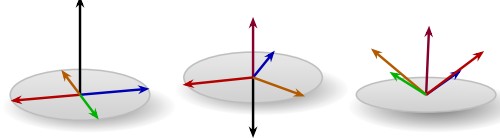

Figure 1: Examples of tight frames with $k = 5$ vectors in $\mathbb{R}^3$.

where $\langle \cdot, \cdot \rangle$ is the inner product in $\mathcal{H}^d$, and $A$ and $B$ are known as the frame bounds. Typically, $k \geq d$ allows the frame to represent vectors with redundancy. An example is illustrated in Figure 1 with $k = 5$ vectors in $\mathbb{R}^3$. We can project any vector in $\mathbb{R}^3$ on these frame vectors without distorting the signal (Casazza & Kutyniok, 2012). Throughout the paper, we operate in real-valued Euclidean spaces $\mathbb{R}^d$ instead of a more general Hilbert space for simplicity.

Fusion frames extend this idea to settings where subspace decomposition is also desired (instead of redundancy alone). Given a collection of subspaces $\{\mathcal{W}_i\}_{i=1}^k$ in $\mathbb{R}^d$ and a corresponding set of

positive weights $\{w_i\}_{i=1}^k$, the collection $(\mathcal{W}_i, w_i)_{i=1}^k$ forms a fusion frame if there exist constants $0 < A \le B < \infty$ such that for all $x \in \mathbb{R}^d$,

$$A||x||^2 \le \sum_{i=1}^k w_i^2 ||P_i x||^2 \le B||x||^2,$$

where $P_i$ denotes the orthogonal projection onto the subspace $\mathcal{W}_i$. The weights $w_i$ adjust the influence of each subspace, and $A$, $B$ are the fusion frame bounds. A fusion frame is tight if $A = B$, and Parseval if $A = B = 1$. Throughout the paper, we focus on Parseval fusion frames. If all weights $w_i$ are equal to 1, we write the fusion frame simply as $\{\mathcal{W}_i\}_{i=1}^k$.

## 2.1 Operators in Fusion Frames

Let $((\mathcal{W}_i, w_i))_{i=1}^k$ be a fusion frame for $\mathbb{R}^d$, with orthonormal basis $(P_i)_{i=1}^k$ respectively. We define three key operators:

The analysis operator $\mathcal{T}_\mathcal{W} : \mathbb{R}^d \to \bigoplus_{i=1}^k \mathcal{W}_i$ maps a vector to its projections across all subspaces:

$$\mathcal{T}_\mathcal{W}(x) = (w_i P_i^T x)_{i=1}^k. \tag{2}$$

The synthesis operator $\mathcal{T}_\mathcal{W}^* : \bigoplus_{i=1}^k \mathcal{W}_i \to \mathbb{R}^d$ re-builds a vector from its fusion frame representation:

$$\mathcal{T}_\mathcal{W}^*((y_i)_{i=1}^k) = \sum_{i=1}^k w_i P_i y_i. \tag{3}$$

The fusion frame operator is the composition of the two operators above, $\mathcal{S}_\mathcal{W} = \mathcal{T}_\mathcal{W}^* \mathcal{T}_\mathcal{W}$, defined by:

$$\mathcal{S}_\mathcal{W}(x) = \sum_{i=1}^k w_i^2 P_i P_i^T x. \tag{4}$$

This operator is self-adjoint, positive semi-definite, and bounded. For Parseval fusion frames, we have $\mathcal{S}_\mathcal{W} = I_d$, allowing exact reconstruction. These operators help in mapping vectors in $\mathbb{R}^d$ to their fusion frame representations and back, and are useful when defining our method.

## 3 Fine-tuning in Fusion Frame Subspaces

We start by analyzing the subspace structure of LoRA (Hu et al., 2022), a particularly effective PEFT method for fine-tuning large pre-trained models. The key idea in LoRA is representing weight updates through a low-rank decomposition. For a layer $l$ with pre-trained parameters $W_l \in \mathbb{R}^{m \times n}$, LoRA decomposes the weight update $\delta W_l$ as a product of two rank-deficient matrices:

$$W_l' = W_l + \delta W_l = W_l + B_l A_l \tag{5}$$

where $B_l \in \mathbb{R}^{m \times r}$, $A_l \in \mathbb{R}^{r \times n}$ and $r < \min(m, n)$.

To better understand the structure of these updates, we analyze the Singular Value Decomposition (SVD) of $\delta W_l$:

$$\delta W_l = U \Sigma V^T = U_r \Sigma_r V_r^T \tag{6}$$

where $U \in \mathbb{R}^{m \times m}$ and $V \in \mathbb{R}^{n \times n}$ are orthogonal matrices spanning the *output* and *input* spaces respectively, and $\Sigma \in \mathbb{R}^{m \times n}$ is a rectangular diagonal matrix. Since the rank of $\delta W_l$ is constrained to $r$, we can simplify this product to $U_r \Sigma_r V_r^T$, where $\Sigma_r$ contains only the top $r$ singular values and $U_r$, $V_r$ are the corresponding left and right singular vectors. This decomposition gives some insight about LoRA updates: $V_r$ defines a subspace in the *input space* $\mathbb{R}^n$ where the input is first projected. Then, $\Sigma_r$ scales these projections, and $U_r$ maps them to a subspace of the *output space* $\mathbb{R}^m$. These input and output subspaces emerge *implicitly* as a by-product of training the matrices $B$ and $A$, without direct control over their properties or interactions.

Recent results indicate that the contribution of matrices $A$ and $B$ is *asymmetric*, which can lead to instability (Hayou et al., 2024), or failure to converge to the optimal solution (Malinovsky et al.,

Figure 2: (*left*) How parameter update $\delta W$ is modeled by LoRA and by FrameFT. (*right*) The parameter update rule in FrameFT. We freeze the Fusion Frame projection matrices and train only the coefficients $C_l$.

2024). So, we can identify two promising opportunities. **(i)** can we explicitly choose and work with multiple subspaces in both input space $\mathbb{R}^n$ and output space $\mathbb{R}^m$ to better capture parameter updates? **(ii)** can we maintain precise control over how these subspaces interact during training? We will demonstrate that Fusion frames provide a framework to address **both questions**, allowing us to decompose both input and output spaces into multiple, potentially overlapping subspaces while controlling their relationships.

### 3.1 Fusion Frames for Finetuning Weight Updates

For the first problem, we want to decompose the input and output spaces into multiple subspaces. The Fusion Frame operators defined in Section 2.1 can be used for this purpose. In particular, let $(P_{n,i}^T)_{i=1}^k$ and $(P_{m,i}^T)_{i=1}^k$ be the orthogonal projection matrices for decomposing the input ($\mathbb{R}^n$) and output ($\mathbb{R}^m$) spaces into $k$ subspaces with dimensions $\rho_n$ and $\rho_m$ respectively. Let,

$$P_n = [P_{n,1}; P_{n,2}; \ldots; P_{n,k}] \quad P_m = [P_{m,1}; P_{m,2}; \ldots; P_{m,k}] \tag{7}$$

We can model richer interactions between these subspaces by introducing trainable coefficient matrices, unlike the simple diagonal matrix $\Sigma_r$ in LoRA. Specifically, we define $(C_{l,i})_{i=1}^k$ where each $C_{l,i} \in \mathbb{R}^{\rho_m \times \rho_n}$ encodes the relationships within the $i$-th input and output subspaces. These matrices can be neatly organized into a block diagonal structure: $C_l = \text{diag}(C_{l,1}, C_{l,2}, \ldots, C_{l,k})$ which yields our parameter update equation:

$$W_l' = W_l + \delta W_l = W_l + \frac{\alpha}{\sqrt{mn}} P_m C_l P_n^T$$

Expanding this expression with the subspaces, we get:

$$W_l' = W_l + \frac{\alpha}{\sqrt{mn}} \sum_{i=1}^k P_{m,i} C_{l,i} P_{n,i}^T$$

Here, $\alpha$ denotes a scaling hyperparameter, normalized by $\sqrt{mn}$ (for dimension independence). Figure 2 (*left*) shows the difference between the SVD decomposition in LoRA and our method. The projection matrices $P_n$ and $P_m$ are obtained from Tight Fusion Frames (TFF) as in §3.2. Under certain conditions, these TFFs exhibit Equichordal and Equi-Isoclinic properties (Fickus et al., 2023b), maximizing subspace separation for efficient representations. These TFFs remain **fixed** in our method, and **only the coefficient matrices** $(C_{l,i})_{i=1}^k$ are trainable. Figure 2 (*right*) shows the FrameFT update mechanism for a single layer (also see Alg. 1).

### 3.2 Constructing Tight Fusion Frames

To keep our construction simple, we consider uniform tight fusion frames, where all subspaces have the same dimension. In practice, one can easily use different subspace dimensions across layers. To construct a $(k, \rho, d)$ uniform tight fusion frame—where $k$ is the number of subspaces, $\rho$ is the dimension of each subspace, and $d$ is the dimension of the entire space—we use the Spectral Tetris algorithm described in Casazza et al. (2011) which is summarised as: (1) Construct a unit norm tight frame (UNTF) in $\mathbb{C}^\rho$ with $d$ vectors. (2) Modulate these vectors using the $k^{\text{th}}$ roots of unity to form $k$ subspaces of dimension $\rho$ in $\mathbb{C}^d$. (3) Use the method in Fickus et al. (2023a) to extend the result to real-valued spaces. This process ensures that the resultant set satisfies the tight fusion frame conditions. An example is included in Appendix §G.

---

**Algorithm 1** FrameFT: Finetuning with Frames

---

**Require:** Frozen Parameters $W_l$, Fusion Frame number of subspaces $k$, subspace dimensions $\rho_m, \rho_n$, number of non-zero coefficients $n_c$, non-zero coefficients $G$, scalar $\alpha$

1: $m,n$ = shape($W_l$)  // get the shape of the parameter matrix
2: $P_n$, $P_m$ = TFF($k, \rho_m, m$), TFF($k, \rho_n, n$)  // generate the TFFs
3: Coeffs = Init(m, n, size=$n_c$)  // initialize the coefficients matrix
4: $C_l$ = rearrange(Coeffs, $m, n, G$)  // re-arrange the coefficients in dense format
5: $\delta W_l = \frac{\alpha}{\sqrt{mn}} P_m C_l P_n^T$  //compute the update term
6: $W_l^{'} = W_l + \delta W_l$  // update the parameter matrix

---

### 3.3 STRUCTURED COEFFICIENT MATRIX

Based on Sparse Fine-Tuning methods (Ansell et al., 2022), we can boost parameter efficiency by introducing sparsity into the coefficient matrices $(C_{l,i})_{i=1}^k$. This sparsification nicely complements our fusion frame architecture: while fusion frames provide *structured subspace decompositions of the parameter space*, sparse coefficients identify the *most useful subspace interactions* (see off-diagonal terms in see Fig. 2). Note that the Fusion Frames themselves are sparse, which is separate from the sparsity in the coefficient matrix, which can be exploited in the implementation. Our experiments indicate that a simple random sparsity pattern performs well: we randomly designate a small subset of entries in $C_l$ as non-zero, distributing them uniformly across the block-diagonal matrices $(C_{l,i})_{i=1}^k$.

This reduces the memory requirements from $O(k\rho_m\rho_n)$ to $O(s)$ where $s \ll k\rho_m\rho_n$. To reduce memory overhead even further, we share these sparsity patterns across layers, i.e., $\text{supp}(C_l) = \text{supp}(C_{l'})$ for layers $l$ and $l'$, where $\text{supp}(\cdot)$ represents the support of a matrix. The cross-layer sharing of sparsity patterns not only reduces memory requirements but also suggests that subspace interactions necessary for task adaptation may share structural patterns across network layers.

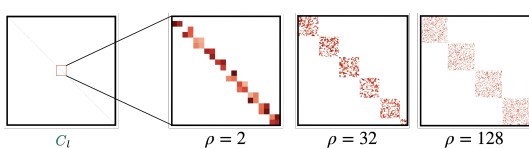

Figure 3: An example of the coefficient matrix $C_l$ from a Query layer (RoBERTa-L) with different subspace dimensions $\rho_m = \rho_n = \rho$.

Figure 3 shows this sparse structure with an example from RoBERTa-Large.

### 3.4 ANALYSIS OF FRAMEFT CONVERGENCE PROPERTIES

Our formulation of the parameter updates using Tight Fusion Frames and Sparse coefficients enjoys theoretical benefits compared to LoRA. Sun et al. (2024b) describes how, for Lipschitz-smooth loss functions, LoRA can create a non-smooth landscape when projected onto the parameters in $A$ and $B$. Through Lemma 1 we prove that FrameFT preserves Lipschitz-smoothness: when a function exhibits Lipschitz smoothness w.r.t. $W$, this property holds when remapped onto our coefficient space $C_l$.

**Lemma 1.** *For any differentiable function $f(W)$ that is L-Lipschitz smooth under Frobenius norm, with fusion frames characterized by frame bounds $(A_m, B_m)$ and $(A_n, B_n)$ for output and input spaces respectively, the transformed function $f(C_l) = f(W_0 + P_m C_l P_n^T)$ obeys:*

$$||\nabla f(C_l^1) - \nabla f(C_l^2)||_F \leq L\sqrt{r}B_m B_n ||C_l^1 - C_l^2||_F$$

*where $r$ represents the rank of the difference matrix $C_l^1 - C_l^2$.*

Since FrameFT preserves smoothness, we can immediately invoke a broad set of convergence results for Lipschitz-smooth functions (Bubeck, 2015; Zhou & Cong, 2017). We highlight one result: gradient descent constrained to learning rates $\eta \leq 1/(2\tilde{L})$ achieves the expected $1/T$ convergence rate to saddle points, as described in Theorem 3.1. Additional details are provided in the Appendix.

**Theorem 3.1.** *Consider minimizing $f(C_l) = f(W_0 + P_m C_l P_n^T)$ where: $f(C_l)$ is $\tilde{L}$-Lipschitz smooth (but potentially non-convex) and $f(C_l)$ is lower bounded by $f^*$. We will use gradient descent with step size $\eta$. Then for step size $\eta \leq 1/(2\tilde{L})$, running gradient descent for $T$ iterations satisfies:*

$$(1/T) \sum_{t=0}^{T-1} ||\nabla f(C_l^t)||_F^2 \leq 2(f(C_l^0) - f^*)/(\eta T)$$

Table 1: Fine-Tuning RoBERTa Base and Large models on GLUE benchmark. * indicates the results reported in prior work. FrameFT performs better than full fine-tuning and LoRA, using $10\times$ fewer parameters.

| Model | Method | Params | SST-2 | MRPC | CoLA | QNLI | RTE | STS-B | Avg. |
|---|---|---|---|---|---|---|---|---|---|
| RoBERTa Base | FF* | 125M | 94.8 | 90.2 | 63.6 | **92.8** | 78.7 | 91.2 | 85.2 |
| | LoRA* | 0.3M | **95.1** | 89.7 | 63.4 | 93.3 | 78.4 | **91.5** | 85.2 |
| | AdaLoRA* | 0.3M | 94.5 | 88.7 | 62.0 | 93.1 | **81.0** | 90.5 | 85.0 |
| | FourierFT* | 24K | 94.2 | 90.0 | 63.8 | 92.2 | 79.1 | 90.8 | 85.0 |
| | SVFit* | 18K | 92.4 | 90.0 | 63.8 | 90.8 | 78.0 | 92.4 | 85.1 |
| | FrameFT (*ours*) | 24K | 94.3 | **92.3** | **66.8** | 92.4 | 79.8 | 90.9 | **86.1** |
| RoBERTa Large | FF* | 356M | **96.4** | 90.9 | 68.0 | 94.7 | 86.6 | **92.4** | 88.2 |
| | LoRA* | 0.8M | 96.2 | 90.2 | 68.2 | **94.8** | 85.2 | 92.3 | 88.2 |
| | RoseLoRA* | 53.4K | 95.2 | 90.2 | 69.2 | 94.7 | 89.2 | 92.0 | 88.5 |
| | FourierFT* | 48K | 96.0 | 90.9 | 67.1 | 94.4 | 87.4 | 91.9 | 88.0 |
| | SVFit* | 36K | 96.2 | 90.9 | **71.4** | 94.4 | 86.3 | 92.0 | 88.5 |
| | FrameFT (*ours*) | 48K | 96.2 | **92.6** | 69.8 | 93.4 | **88.1** | 91.9 | **88.7** |

## 4 EXPERIMENTS

In this section, we compare the performance of Transformer-based models fine-tuned with FrameFT. All our experiments are conducted on two NVIDIA A100 GPUs with 40GB of memory each. Individual sections include more details on the experimental setup. Additional experiments with varying hyperparameters are available in the appendix.

### 4.1 NATURAL LANGUAGE UNDERSTANDING CAPABILITIES

**Evaluation Framework:** We evaluate the performance of FrameFT by fine-tuning the base and large variants of RoBERTa Liu et al. (2019) across multiple GLUE benchmark tasks Wang et al. (2019). This suite of benchmarks covers sentiment classification, paraphrase detection, and entailment recognition and provides a standardized setup for testing.

**Fine-tuning strategy:** We use the original LoRA Hu et al. (2022) recipe by finetuning the Query and Value matrices across network layers. For FrameFT, we used 1000 non-zero coefficient entries with their positions determined randomly and shared across all layers. For the tight fusion frame construction, we use a subspace dimension $\rho = 2$ and subspace count $k$ was calibrated so that $k\rho = n$ ($n$ is dimension of the layer).

**Performance analysis:** We report the Pearson correlation coefficient (PCC) for STS-B task, Matthew's correlation coefficient (MCC) for CoLA and accuracy for the remaining tasks. For the baseline methods, we report LoRA Hu et al. (2022), AdaLoRA Zhang et al. (2023), FourierFT Gao et al. (2024), SVFit Sun et al. (2024a) and RoseLoRA Wang et al. (2024). The results are presented in Table 4.1. We see that FrameFT performs on par or better than LoRA and full fine-tuning on individual tasks, and performs better than both methods on average. FrameFT achieves this with $10\times$ fewer parameters when compared to LoRA. We also note that even though the number of parameters of SVFit Sun et al. (2024a) is slightly lower than FrameFT, SVFit trains the singular values, keeping the singular vectors fixed. So, in practice, one would need to save the singular vectors for each layer after training, which increases the storage requirements. Also, for RoseLoRA Wang et al. (2024), only the parameter count is reported in Table 4.1. but they use a different mask for each layer, so the storage cost is $3\times$ the number of parameters if we account for the location of the non-zero coefficients.

### 4.2 INSTRUCTION TUNING

**Benchmarking Framework:** We evaluate FrameFT for fine-tuning LLMs to follow instructions. We finetune the 7B and 13B variants of Llama2 models Touvron et al. (2023), Gemma2 models Team et al. (2024b) with 2B and 9B parameters and Llama 3.1 Grattafiori et al. (2024) 8B model on the Alpaca instruction dataset Taori et al. (2023). We evaluate the performance of the fine-tuned

models on the LM-evaluation harness Gao et al. (2023) by Eleuther AI. We incorporate eight distinct challenge categories spanning reasoning, world knowledge, and generalization capabilities.

**FrameFT configuration:** We apply FrameFT with $n = 5000$ non-zero coefficients and adapt the Query and Value matrices for all the Transformer blocks. We determine the position of the non-zero coefficients at random and share them across all the layers. We use $\alpha = 200$ across all of our experiments. We set the subspace dimension $\rho = 2$ and as before, calculate the number of subspaces $k$ such that $k\rho = n$ for each layer. For all baselines, we choose the hyperparameters suggested for the respective method.

**Performance Analysis:** We compare FrameFT with LoRA Hu et al. (2022), (IA)$^3$ Liu et al. (2022a), DoRA Liu et al. (2024), $S^2FT$ Yang et al. (2024) and FourierFT Gao et al. (2024). Our results are shown in Table 3. We observe that FrameFT performs on par or better compared to the baselines while using the fewest number of parameters across all models. In addition, FrameFT also provides computational benefits as described in Section 4.4.

### 4.3 PERFORMANCE ON VISION TRANSFORMER MODELS

We performed a set of experiments to determine whether FrameFT was effective only for Language models? To check this, we applied FrameFT to fine-tune Vision Transformers on 8 image classification tasks, which include remote sensing, fine-grained classification, and texture recognition.

**Performance Analysis:** Appendix Section C shows the performance of FrameFT across these tasks for ViT-L and ViT-B models. We observe that FrameFT performs better than the baseline methods (which include full-finetuning) on average. Moreover, we observe performance improvements across the majority of the tasks when we increase the number of parameters for FrameFT. These results indicate that FrameFT generalizes well across both Vision and Language models. More experimental details are presented in Appendix C and Table 6.

### 4.4 OPERATIONAL EFFICIENCY/LATENCY ANALYSIS

In this section, we analyze the throughput of FrameFT measured in tokens per second. In practice, when serving LLMs to different users, the base model is held fixed, and various specialized variants are maintained simultaneously. Hence, we measure the throughput of the adapter layer introduced by different PEFT methods, with the idea that the base layer throughput remains the same in the pretrained and fine-tuned models.

**Performance Analysis:** Table 4.4 displays the tokens per second for various methods across different model families. All measurements are performed on an NVIDIA A100 machine. We observe that FrameFT performs better than LoRA, which in turn performs better than FourierFT and other methods. This highlights FrameFT's efficiency benefits due to sparse coefficients and sparse projection matrices as described in §3.3 and Appendix §G

Table 2: Tokens per second for different PEFT methods across different models. FrameFT shows higher throughput owing to the sparsity in the coefficient matrix and the Fusion Frame projection matrices.

| Model | Method | #params | #tokens/sec |
|---|---|---|---|
| | LoRA | 262k | 23.6k |
| | DoRA | 262k | 14.1k |
| Llama-2-7b | SVFit | 1k | 9.0k |
| | FourierFT | 1k | 1.8k |
| | FrameFT (*ours*) | 1k | **39.4$k$** |
| | LoRA | 245k | 22.1k |
| | DoRA | 245k | 16.0k |
| Gemma-2-9b | SVFit | 1k | 12.4 |
| | FourierFT | 1k | 1.4k |
| | FrameFT (*ours*) | 1k | **29.6$k$** |
| | LoRA | 163k | 22.3k |
| | DoRA | 163k | 18.7k |
| Llama-3.1-8b | SVFit | 1k | 11.5k |
| | FourierFT | 1k | 1.6k |
| | FrameFT (*ours*) | 1k | **27.3$k$** |

We note that constructing the Fusion Frames takes a finite amount of time. Specifically, the time complexity for constructing a $(k, \rho, d)$ TFF is $\mathcal{O}(kd)$. However, this generation step is performed only *once* during initialization, and the resulting projection matrices are shared across all network layers, amortizing the one-time compute cost across multiple layers and forward calls. So, we focus on per-layer throughput measurements.

### 4.5 PERFORMANCE AS A FUNCTION OF NUMBER OF NON-ZERO COEFFICIENTS

We evaluate the performance of FrameFT as we increase the number of non-zero coefficients. We choose the RoBERTa base model for this experiment and check the performance of the model fine-tuned with FrameFT on different tasks in the GLUE benchmark. Figure 4 shows expected trends:

Table 3: Performance of LLMs fine-tuned on the Alpaca dataset by various methods and then evaluated on the LM-evaluation-harness. FrameFT performs competitively with all the baseline methods under comparison, using a $30\times$ fewer number of parameters than LoRA. FrameFT needs minimal hyperparameter tuning.

| Model | Method | #Params | ARC-c | ARC-e | BoolQ | HellaSwag | OBQA | PIQA | RTE | WinoGrande | Avg. |
|---|---|---|---|---|---|---|---|---|---|---|---|
| Llama-2-7b | LoRA | 16.7M | 45.82 | 77.02 | 78.81 | 58.08 | 35.2 | 78.83 | 61.01 | 70.72 | 63.18 |
| | $(IA)^3$ | 614K | 45.38 | 76.60 | 77.89 | 57.74 | 34.6 | 78.78 | 64.26 | 68.82 | 63.01 |
| | DoRA | 16.7M | 45.14 | 76.39 | 78.35 | 57.78 | 34.0 | 78.73 | 67.15 | 67.17 | 63.09 |
| | $S^2FT$ | 56.6M | 46.16 | 77.44 | 78.38 | 58.10 | 32.8 | 78.84 | 61.01 | 68.58 | 62.66 |
| | FourierFT | 320K | 44.96 | 77.14 | 79.05 | 58.21 | 34.6 | 78.89 | 62.45 | 70.48 | 63.22 |
| | FrameFT (*ours*) | 320K | 45.22 | 76.93 | 78.62 | 58.08 | 34.2 | 78.62 | 66.06 | 71.19 | **63.62** |
| Llama-2-13b | LoRA | 26.2M | 50.77 | 80.35 | 81.44 | 61.13 | 36.0 | 79.38 | 69.68 | 72.61 | 66.42 |
| | $(IA)^3$ | 963K | 50.25 | 79.96 | 80.52 | 60.54 | 34.2 | 79.27 | 68.59 | 72.22 | 65.69 |
| | DoRA | 26.2M | 51.79 | 80.13 | 80.21 | 61.44 | 35.6 | 79.81 | 71.48 | 72.21 | **66.58** |
| | $S^2FT$ | 111M | 20.73 | 35.10 | 46.17 | 32.94 | 14.6 | 58.48 | 55.23 | 50.04 | 39.16 |
| | FourierFT | 400K | 49.74 | 80.17 | 80.82 | 60.98 | 35.6 | 79.76 | 64.62 | 72.06 | 65.47 |
| | FrameFT (*ours*) | 400K | 50.34 | 79.75 | 81.19 | 60.87 | 35.8 | 80.08 | 68.59 | 72.29 | 66.11 |
| Gemma-2-2B | LoRA | 6.4M | 48.63 | 79.76 | 76.61 | 55.85 | 31.6 | 78.94 | 62.09 | 68.19 | 62.70 |
| | $(IA)^3$ | 292K | 46.76 | 80.13 | 70.70 | 55.79 | 31.2 | 78.13 | 59.21 | 69.77 | 61.46 |
| | DoRA | 6.4M | 48.72 | 80.35 | 72.26 | 55.96 | 33.6 | 78.56 | 67.15 | 68.98 | 63.20 |
| | $S^2FT$ | 16.9M | 45.90 | 78.41 | 59.24 | 54.80 | 31.4 | 77.80 | 57.40 | 56.27 | 57.65 |
| | FourierFT | 260K | 45.98 | 79.12 | 73.88 | 55.57 | 32.8 | 79.16 | 67.87 | 67.87 | 62.78 |
| | FrameFT (*ours*) | 260K | 48.37 | 81.06 | 75.13 | 55.71 | 34.6 | 79.05 | 69.67 | 69.30 | **64.11** |
| Gemma-2-9B | LoRA | 17.9M | 64.76 | 88.22 | 86.39 | 62.96 | 36.2 | 82.48 | 70.40 | 75.30 | **70.83** |
| | $(IA)^3$ | 774K | 61.95 | 87.21 | 85.05 | 62.11 | 35.8 | 81.56 | 68.95 | 74.27 | 69.61 |
| | DoRA | 17.9M | 62.62 | 87.16 | 86.02 | 62.91 | 35.2 | 81.61 | 71.48 | 73.71 | 69.09 |
| | $S^2FT$ | 74.4M | 53.41 | 80.55 | 80.73 | 59.02 | 32.8 | 79.54 | 69.67 | 70.32 | 65.76 |
| | FourierFT | 420K | 64.16 | 88.21 | 86.36 | 62.78 | 36.4 | 81.66 | 70.03 | 74.27 | 70.48 |
| | FrameFT (*ours*) | 420K | 65.01 | 88.00 | 86.02 | 63.09 | 37.2 | 81.72 | 69.31 | 74.82 | 70.65 |
| Llama-3.1-8B | LoRA | 13.6M | 55.38 | 83.38 | 82.08 | 61.73 | 35.2 | 81.23 | 74.01 | 75.53 | **68.56** |
| | $(IA)^3$ | 524K | 54.27 | 82.79 | 82.20 | 61.04 | 34.4 | 80.58 | 68.95 | 74.59 | 67.35 |
| | DoRA | 13.6M | 54.01 | 82.07 | 81.31 | 61.07 | 34.6 | 81.23 | 67.51 | 72.77 | 66.82 |
| | $S^2FT$ | 65.4M | 31.57 | 60.02 | 59.82 | 44.24 | 20.0 | 68.88 | 56.68 | 53.67 | 49.36 |
| | FourierFT | 320K | 51.36 | 80.51 | 81.49 | 60.57 | 34.0 | 80.41 | 70.39 | 73.95 | 66.58 |
| | FrameFT (*ours*) | 320K | 53.41 | 82.53 | 82.29 | 60.79 | 34.6 | 80.90 | 75.09 | 73.40 | 67.87 |

the performance of the fine-tuned model on different tasks increases as we increase the number of non-zero terms in the coefficient matrix. Table C also confirms this behavior as we increase $n$ from 1000 to 5000. These results show that FrameFT is able to utilize the additional parameters to improve the fine-tuning performance for both vision and language tasks. Additional experiments on more tasks from the GLUE benchmark are included in appendix E.

## 5 RELATED WORK

Model adaptation for downstream tasks is being studied extensively in recent years, and has provided various efficient methods that reduce computation and storage needs while maintaining performance. Here, we describe different variants of PEFT methods briefly introduced in § 1.

**Adapters:** Adapters introduce specialized modules between pre-existing layers within the pretrained model. These adapter layers are trained during fine-tuning while keeping the pretrained model parameters frozen (Houlsby et al., 2019; Karimi Mahabadi et al., 2021; He et al., 2022). By keeping the parameters of the original model frozen, they preserve the knowledge acquired during pretraining while reducing the risk of overfitting.

**Low-Rank Matrix Factorization:** LoRA techniques reparameterize the weight updates on selected layers of the model through low-rank factorizations (Hu et al., 2022). This framework of training only the decomposition matrices while freezing pretrained parameters has led to many variants exploring asymmetric chaining (Malinovsky et al., 2024), quantization-aware formulations (Dettmers et al., 2023), and different learning rates for the update terms (Hayou et al., 2024).

**Prefix Tuning:** Prefix-tuning strategies prepend learnable vector sequences before transformer layer inputs, creating controlable input modifications (Wang et al., 2025; Li & Liang, 2021; Qin & Eisner, 2021). These prefix vectors are adapted to the downstream task while weights of the pretrained model are fixed (Lester et al., 2021b; Liu et al., 2022b).

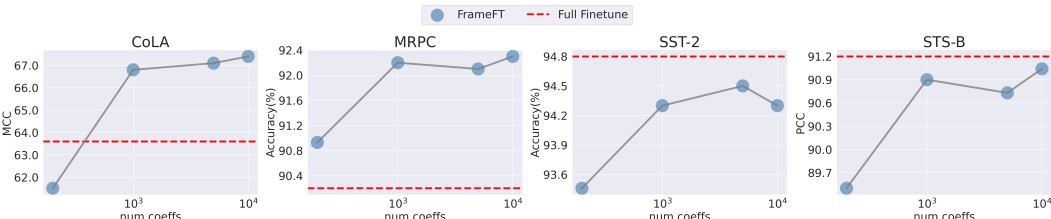

Figure 4: Performance of RoBERTa base model fine-tuned with FrameFT versus the number of non-zero coefficients. The red line indicates the performance of full fine-tuning.

**Prompt Tuning:** Prompt-tuning approaches learn different prompt representations that guide model behaviors without architectural modification (Xiao et al., 2025; Lester et al., 2021a; Liu et al., 2022c; Ge et al., 2022). Unlike prefix tuning, prompt tuning operates exclusively at the input embedding level, making it particularly efficient and easy to integrate (Lester et al., 2021a). Prompt tuning has also been extended to domain adaptation (Ge et al., 2022), vision-language models (Zhou et al., 2022), and diffusion models (Dong et al., 2023; Chung et al., 2023).

**Sparse Fine-Tuning:** Sparse fine-tuning methodologies exploit natural parameter redundancy by targeting only critical components while freezing the rest (Khaki et al., 2025; Lu et al., 2024; Gao et al., 2024; Guo et al., 2021). These approaches – whether through low-rank operations Lu et al. (2024) or spectral compression via Fourier transformations Gao et al. (2024) – achieve competitive or superior performance compared to full fine-tuning while reducing trainable parameter count.

## 6 DISCUSSIONS

We cover a few relevant points not discussed in detail so far. **(a)** *Fine-tuning versus prompt-tuning?* Previous works have studied the pros/cons of prompt tuning compared to PEFT methods Wistuba et al. (2024); Pu et al. (2023). In particular, Pu et al. (2023) shows a quantitative analysis of PEFT methods such as LoRA and $(IA)^3$ versus prompt tuning and reports that PEFT methods perform better than prompt tuning in general. Since FrameFT performs on par or better than LoRA, we expect the results to translate to prompt-tuning as well. **(b)** *Frames instead of Fusion Frames?* Indeed, Frames (subspace dimension = 1) can be used instead of Fusion Frames. The construction we outline in §3.2 has a minimum subspace dimension of 2 since we first construct a Fusion Frame for the Complex domain and then transform it to the Real domain. If a subspace dimension of 1 (while keeping $k\rho = n$) is indeed required, one could use any orthonormal basis, such as a Fourier basis or Wavelets. **(c)** *How does performance vary as a function of subspace dimension?* We performed experiments changing the subspace dimension and report the results in §D. We observe that as the subspace dimension is increased (with $k\rho = n$), the sparsity in Fusion Frames increases, thereby reducing the degrees of freedom for the parameter update. This leads to a small drop in performance.

## 7 CONCLUSIONS

We describe FrameFT, a parameter-efficient fine-tuning framework that leverages structured subspace decompositions based on Fusion Frames to fine-tune transformer models for vision and language tasks. Our extensive empirical validation across both vision transformers and state-of-the-art language models (including the Llama and Gemma families), shows that substantial compute and parameter efficiency gains are achievable without sacrificing performance across many evaluation benchmarks. We also provide a technical result showing that FrameFT preserves the Lipschitz smoothness of the loss landscape, and therefore achieves desirable convergence properties. We believe that one strong advantage of parameter efficiency of FrameFT will be in situations where the use case requires a *set* of fine-tuned models, each fine-tuned on a specific task – these are invoked on a case by case basis. We note that support for structured sparsity (beyond $2:4$ sparsity) remains limited but this provides a concrete opportunity for higher efficiency gains if specialized kernels are implemented.

## 8 REPRODUCIBILITY STATEMENT

To ensure the reproducibility of our work, all hyperparameters used for the experiments on the GLUE benchmark, image classification tasks, and instruction tuning are provided in Appendix F (Tables 5, 6, and 7). The theoretical analysis, including the proofs for our convergence claims and details on the construction of Tight Fusion Frames, is available in the appendix. The code and experimental setup will be made public upon acceptance of the paper.

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

APPENDIX

In this appendix, we provide proofs for the theorems and additional details on the experiments presented in the main paper. In Section A, we provide a detailed proof for the lemma showing that FrameFT preserves the Lipschitz smoothness of the loss function. Section B presents the results for the convergence guarantee of Gradient Descent to a saddle point for Lipschitz smooth functions. Section C evaluates the performance of FrameFT for Image classification tasks. In Section D, we measure the performance of FrameFT as we vary the subspace dimension. In Section E, we show the performance of FrameFT with an increasing number of non-zero coefficients. We observe that FrameFT can leverage the additional parameters to improve the performance. Section F lists the hyperparameters used for our experiments. Finally, in Section G, we describe an algorithm for constructing Tight Fusion Frames along with an example.

## A FRAMEFT PRESERVES LIPSCHITZ SMOOTHNESS

Here, for working through the proof, assume the following dimensions for the matrices involved: $P_l$ is $m \times p$, $P_{prev}$ is $n \times q$, $C_l$ is $p \times q$ and $W_0$ is $m \times n$.

**Proof.** Let us consider how the loss function behaves at two different points. Take any two coefficient matrices $C_l^1$ and $C_l^2$. From our composition rule, these map to: $W^1 = W_0 + P_l C_l^1 P_{prev}^T$ and $W^2 = W_0 + P_l C_l^2 P_{prev}^T$.

By the $L$-Lipschitz smoothness assumption on $f(W)$:
$$||\nabla f(W^1) - \nabla f(W^2)||_F \leq L||W^1 - W^2||_F$$
where we use the notation $\nabla f(W) = \frac{\partial f}{\partial W}$, the derivative of $f$ with respect to $W$. We also have $W^1 - W^2 = P_l(C_l^1 - C_l^2)P_{prev}^T$ and we will show shortly that $||W^1 - W^2||_F$ is upper-bounded by terms involving $(C_l^1 - C_l^2)$ and other constants. The chain rule for matrix derivatives means the derivative of $f$ with respect to the coefficients $C_l$ is,
$$\nabla f(C_l) = \frac{\partial f}{\partial W} \cdot \frac{\partial W}{\partial C_l} = \nabla f(W) \cdot \nabla W(C_l).$$

The term $\nabla W(C_l)$ denotes how the weight matrix $W$ changes with respect to a change in the coefficient matrix $C_l$. We will use directional derivatives to compute this.

Consider a matrix $\zeta_\Delta \in \mathbb{R}^{k\rho_l \times k\rho_{prev}}$ which represents a direction. The directional derivative of $W$ with respect to $C_l$ in the direction of $\zeta_\Delta$ is given by
$$\nabla_{\zeta_\Delta} W(C_l) = \lim_{t \to 0} \frac{W(C_l + t\zeta_\Delta) - W(C_l)}{t} = \lim_{t \to 0} \frac{P_l \zeta_\Delta P_{prev}^T t}{t} = P_l \zeta_\Delta P_{prev}^T$$
Hence, the derivative of $f$ with respect to $C_l$ in the direction of $\zeta_\Delta$ is given by
$$\begin{aligned}
\nabla_{\zeta_\Delta} f(C_l) &= \nabla f(W) \cdot \nabla_{\zeta_\Delta} W(C_l) \\
&= \text{Tr}((\nabla f(W))^T (P_l \zeta_\Delta P_{prev}^T)) \\
&= \text{Tr}(P_{prev}^T \nabla f(W)^T P_l \zeta_\Delta) \text{ use } \text{Tr}(AB) = \text{Tr}(BA) \\
&= \text{Tr}((P_l^T \nabla f(W) P_{prev})^T \zeta_\Delta) \text{ use } (ABC)^T = C^T B^T A^T \\
&= \langle P_l^T \nabla f(W) P_{prev}, \zeta_\Delta \rangle
\end{aligned}$$
From the definition of the directional gradient, we have that $\nabla_{\zeta_\Delta} f(C_l) = \langle \nabla f(C_l), \zeta_\Delta \rangle$. But from above, we have $\nabla_{\zeta_\Delta} f(C_l) = \text{Tr}((P_l^T \nabla f(W) P_{prev})^T \zeta_\Delta)$. These two expressions are equal for all $\zeta_\Delta$. Therefore,
$$\nabla f(C_l) = P_l^T \nabla f(W) P_{prev}$$

Hence, for two coefficient matrices $C_l^1, C_l^2$:
$$\begin{aligned}
||\nabla f(C_l^1) - \nabla f(C_l^2)||_F &= ||P_l^T (\nabla f(W^1) - \nabla f(W^2)) P_{prev}||_F \\
&\leq ||P_l||_{op} ||(\nabla f(W^1) - \nabla f(W^2)) P_{prev}||_F \\
&\leq ||\nabla f(W^1) - \nabla f(W^2)||_F ||P_l||_{op} ||P_{prev}||_{op} \\
&\leq L||W^1 - W^2||_F \sqrt{B_l} \sqrt{B_{prev}}
\end{aligned} \tag{8}$$

Now, we need to bound $||W^1 - W^2||_F$. We know that $W^1 = W_0 + P_l C_l^1 P_{prev}^T$ and $W^2 = W_0 + P_l C_l^2 P_{prev}^T$. Therefore:

$$W^1 - W^2 = P_l(C_l^1 - C_l^2)P_{prev}^T \tag{9}$$

Let $M = C_l^1 - C_l^2$. Applying SVD on $M$ gives

$$M = \sum_{i=1}^{r} \sigma_i u_i v_i^T$$

where $r$ is the rank of $M$, $\sigma_1 \geq \sigma_2 \geq \ldots \sigma_r \geq 0$ are the singular values of $M$, $(u_i)_{i=1}^{m}, (v_i)_{i=1}^{n}$ are the orthonormal vectors spanning $\mathbb{R}^m, \mathbb{R}^n$ respectively. Substituting the SVD into the expression for $W^1 - W^2$, we get:

$$W^1 - W^2 = \sum_{i=1}^{r} \sigma_i(P_l u_i)(P_{prev} v_i)^T \tag{10}$$

The square of the leftmost Frobenius norm in inequality above becomes

$$||W^1 - W^2||_F^2 = ||P_l M P_{prev}^T||_F^2$$

$$= ||\sum_{i=1}^{r} \sigma_i(P_l u_i)(P_{prev} v_i)^T||_F^2$$

$$= \sum_{i,j=1}^{r} \langle \sigma_i(P_l u_i)(P_{prev} v_i)^T, \sigma_j(P_l u_j)(P_{prev} v_j)^T \rangle$$

In the summation, each term is of the form: $\sigma_i(P_l u_i)(P_{prev} v_i)^T$. So, when we check the inner product of two such terms, corresponding to indices $i$ and $j$, we get:

$$\langle \sigma_i(P_l u_i)(P_{prev}^T v_i)^T, \sigma_j(P_l u_j)(P_{prev} v_j)^T \rangle = \sigma_i \sigma_j \langle (P_l u_i)(P_{prev} v_i)^T, (P_l u_j)(P_{prev} v_j)^T \rangle \tag{11}$$

Using the definition of inner product $\langle A, B \rangle = \text{Tr}(A^T B)$, for the terms of interest, we get

$$\langle (P_l u_i)(P_{prev} v_i)^T, (P_l u_j)(P_{prev} v_j)^T \rangle = \text{Tr}\left( \left[ (P_l u_i)(P_{prev} v_i)^T \right]^T (P_l u_j)(P_{prev} v_j)^T \right)$$

$$\overset{(a)}{=} \text{Tr}\left( (P_{prev} v_i)(P_l u_i)^T (P_l u_j)(P_{prev} v_j)^T \right)$$

$$\overset{(b)}{=} \text{Tr}\left( (P_{prev} v_j)^T (P_{prev} v_i)(P_l u_i)^T (P_l u_j) \right)$$

$$\overset{(c)}{=} \left( (P_{prev} v_j)^T (P_{prev} v_i) \right)\left( (P_l u_i)^T (P_l u_j) \right)$$

$$= \langle (P_{prev} v_j), (P_{prev} v_i) \rangle \langle (P_l u_i), (P_l u_j) \rangle$$

$$\overset{(c)}{\leq} ||P_{prev} v_i|| \cdot ||P_l u_j|| \cdot ||P_{prev} v_j|| \cdot ||P_l u_i||$$

Above, for the equality (a) we use $(AB)^T = B^T A^T$. The equality (b) uses the cyclic property of trace $\text{Tr}(ABCD) = \text{Tr}(BCDA)$. In (c), we perform the matrix multiplication and notice that $(u_i^T P_l^T)(P_l u_j)$ and $(v_j^T P_{prev}^T)(P_{prev} v_i)$ are scalars. Finally, we rewrite the scalar products in terms of inner products and apply Cauchy-Schwarz to each inner product term.

Let us look at one of the norms above. Since $u_i, u_j, v_i, v_j$ are unit vectors, we have

$$||P_l u_i|| \leq ||P_l||_2 ||u_i||$$

$$\leq \sqrt{B_l} \cdot 1$$

$$= \sqrt{B_l}$$

Similarly, we have

$$||P_l u_j|| \leq \sqrt{B_l}, \quad ||P_{prev} u_i|| \leq \sqrt{B_{prev}}, \quad ||P_{prev} u_j|| \leq \sqrt{B_{prev}}$$

Substituting these results back

$$||W^1 - W^2||_F^2 = \sum_{i,j=1}^{r} \sigma_i \sigma_j \langle P_l u_i, P_l u_j \rangle \langle P_{prev} v_i, P_{prev} v_j \rangle$$

$$\leq \sum_{i,j=1}^{r} \sigma_i \sigma_j ||P_l u_i|| \cdot ||P_l u_j|| \cdot ||P_{prev} v_i|| \cdot ||P_{prev} v_j||$$

$$\leq \sum_{i,j=1}^{r} \sigma_i \sigma_j B_l B_{prev}$$

$$= B_l B_{prev} \sum_{i,j=1}^{r} \sigma_i \sigma_j$$

$$= B_l B_{prev} \left( \sum_{i=1}^{r} \sigma_i \right)^2$$

Taking square root,

$$||W^1 - W^2||_F \leq \sqrt{B_l B_{prev}} \sum_{i=1}^{r} \sigma_i \tag{12}$$

Since $\sum_i \sigma_i = ||M||_*$ (nuclear norm) and $||M||_* = \sqrt{r}||M||_F$ where $r$ is the rank of $M$, we have

$$||W^1 - W^2||_F \leq \sqrt{B_l B_{prev}} \sqrt{r} ||M||_F \tag{13}$$

Substituting this bound in (8), we get

$$||\nabla f(C_l^1) - \nabla f(C_l^2)||_F \leq \sqrt{B_l B_{prev}} L \left( \sqrt{r B_l B_{prev}} ||M||_F \right)$$
$$= L\sqrt{r} B_l B_{prev} ||C_l^1 - C_l^2||_F$$

Hence, $f(C_l)$ is $\tilde{L}$-Lipschitz smooth with $\tilde{L} = L\sqrt{r} B_l B_{prev}$.

## B  CONVERGENCE GUARANTEE BASED ON LIPSCHITZ SMOOTHNESS

The gradient descent update at iteration $t$ is: $C_l^{t+1} = C_l^t - \eta \nabla f(C_l^t)$.

Let $\tilde{L}$ be the Lipschitz constants derived in Section A. By $\tilde{L}$-Lipschitz smoothness:

$$f(C_l^{t+1}) \leq f(C_l^t) + \langle \nabla f(C_l^t), C_l^{t+1} - C_l^t \rangle + (\tilde{L}/2)||C_l^{t+1} - C_l^t||_F^2$$

Substituting $C_l^{t+1} = C_l^t - \eta \nabla f(C_l^t)$ we get:

$$f(C_l^{t+1}) \leq f(C_l^t) + \langle \nabla f(C_l^t), -\eta \nabla f(C_l^t) \rangle + (\tilde{L}/2)\eta^2 ||\nabla f(C_l^t)||_F^2 \tag{14}$$

$$= f(C_l^t) - \eta ||\nabla f(C_l^t)||_F^2 + (\tilde{L}/2)\eta^2 ||\nabla f(C_l^t)||_F^2 \tag{15}$$

Progress per step is

$$f(C_l^t) - f(C_l^{t+1}) \geq \eta ||\nabla f(C_l^t)||_F^2 (1 - (\tilde{L}\eta)/2)$$

Summing from $t = 0$ to $T - 1$:

$$\sum_{t=0}^{T-1}[f(C_l^t) - f(C_l^{t+1})] \geq \eta(1 - (\tilde{L}\eta)/2) \sum_{t=0}^{T-1} ||\nabla f(C_l^t)||_F^2$$

The left side telescopes:

$$f(C_l^0) - f(C_l^T) \geq \eta(1 - (\tilde{L}\eta)/2) \sum_{t=0}^{T-1} ||\nabla f(C_l^t)||_F^2$$

Since $f(C_l^T) \geq f^*$, we have:

$$f(C_l^0) - f^* \geq \eta(1 - (\tilde{L}\eta)/2) \sum_{t=0}^{T-1} ||\nabla f(C_l^t)||_F^2$$

Dividing both sides by $T$ and rearranging:

$$(1/T) \sum_{t=0}^{T-1} ||\nabla f(C_l^t)||_F^2 \leq (f(C_l^0) - f^*)/(\eta(1 - (\tilde{L}\eta)/2)T)$$

For $\eta \leq 1/(2\tilde{L})$, we have $1 - (\tilde{L}\eta)/2 \geq 1/2$, and so

$$(1/T) \sum_{t=0}^{T-1} ||\nabla f(C_l^t)||_F^2 \leq 2(f(C_l^0) - f^*)/(\eta T)$$

## C  IMAGE CLASSIFICATION WITH VISION TRANSFORMERS

**Evaluation Framework:** We investigate the effectiveness of FrameFT for fine-tuning Vision Transformers. To this end, we fine-tune the base and large variants of the Vision Transformer (ViT) architecture Dosovitskiy et al. (2021). We chose the ImageNet-21K pre-trained ViT models available on the Hugging Face Hub for our base model and evaluated across a diverse set of image classification challenges. The test suite includes fine-grained classification tasks – Oxford Pets Parkhi et al. (2012), Stanford Cars, FGVC Aircraft Maji et al. (2013), CIFAR10, CIFAR100, texture recognition – DTD Cimpoi et al. (2014), and remote sensing applications – EuroSAT Helber et al. (2018), RESISC45 Cheng et al. (2017).

**FrameFT configuration:** For this experiment, we utilized sparse block diagonal coefficient matrices with $n = 1000$ and $n = 5000$ non-zero elements, and a scaling factor $\alpha = 250$ consistently across all the tasks. We share the positions of non-zero positions across all the layers. Similar to the natural language understanding experiment, we set $\rho = 2$ while adjusting the number of subspaces $k$ to satisfy $k\rho = n$ throughout the net-

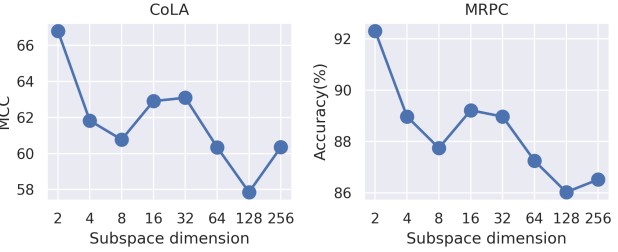

Figure 5: Performance of FrameFT on CoLA and MRPC for different values of subspace dimension $\rho$

work. Again, following LoRA, we adapt only the Query and Value matrices in the self-attention blocks in all layers of the model.

**Performance analysis:** Table C shows the performance of FrameFT for fine-tuning Vision Transformers on diverse datasets. On average, with 5000 non-zero coefficients per layer, FrameFT performs better than all the baseline methods except full fine-tuning. FrameFT is able to achieve this with the lowest number of parameters compared to all the baseline methods, which is $5-20\times$ lower than LoRA. We also see an improvement in performance as we increase the number of non-zero coefficients from 1000 to 5000. As an additional note, the hyperparameters for FrameFT are held fixed for all the tasks in this benchmark. The hyperparameters for FourierFT were adjusted on a task-by-task basis, as noted by Gao et al. (2024).

Table 4: Performance of different methods for Fine-Tuning ViT Base and Large models on different datasets. * indicates the results reported in previous work. We highlight non PEFT methods in gray. FrameFT performs better than all the baseline methods on average, using a 30x lower number of parameters than LoRA.

| | Method | Params | Pets | Cars | CIFAR10 | DTD | EuroSAT | FGVC | RESISC45 | CIFAR100 | Avg. |
|---|---|---|---|---|---|---|---|---|---|---|---|
| ViT-B | Full finetune* | 85.8M | 93.14 | 79.78 | 98.92 | 77.68 | 99.05 | 54.84 | 96.13 | 92.38 | 86.49 |
| | LinearProbe* | – | 90.28 | 25.76 | 96.41 | 69.77 | 88.72 | 17.44 | 74.22 | 84.28 | 68.36 |
| | LoRA* | 581K | 93.19 | 45.38 | **98.78** | 74.95 | 98.44 | 25.16 | 92.70 | **92.02** | 77.58 |
| | FourierFT | 72K | 93.21 | 46.11 | 98.58 | 75.09 | 98.29 | 27.51 | 91.97 | 91.20 | 77.75 |
| | FourierFT | 239K | 93.05 | 56.36 | 98.69 | 77.30 | 98.78 | 32.44 | **94.26** | 91.45 | 80.29 |
| | FrameFT (*ours*) | 24K | **93.71** | 70.84 | 98.50 | 77.80 | 98.52 | 44.45 | 92.30 | 90.80 | 83.36 |
| | FrameFT (*ours*) | 120K | 93.55 | **78.16** | 98.50 | **79.96** | 98.91 | 52.35 | 94.10 | 90.90 | **85.80** |
| ViT-L | Full finetune* | 303.3M | 94.43 | 88.90 | 99.15 | 81.79 | 99.04 | 68.25 | 96.43 | 93.58 | 90.20 |
| | LinearProbe* | – | 91.11 | 37.91 | 97.78 | 73.33 | 92.64 | 24.62 | 82.02 | 84.28 | 72.96 |
| | LoRA* | 1.57M | 94.82 | 73.25 | **99.13** | 81.79 | 98.63 | 42.32 | 94.71 | **94.87** | 84.94 |
| | FourierFT | 144K | 94.46 | 69.56 | 99.10 | 80.83 | 98.65 | 39.92 | 93.86 | 93.31 | 83.71 |
| | FourierFT | 480K | 94.84 | 79.14 | 99.08 | **81.88** | 98.66 | 51.28 | **95.20** | 93.37 | 86.68 |
| | FrameFT (*ours*) | 48K | 94.27 | 78.48 | 99.02 | 79.94 | 98.49 | 52.63 | 93.72 | 92.68 | 86.15 |
| | FrameFT (*ours*) | 240K | 94.20 | **82.83** | 98.90 | 81.54 | **98.87** | 59.63 | 94.96 | 92.71 | **87.95** |

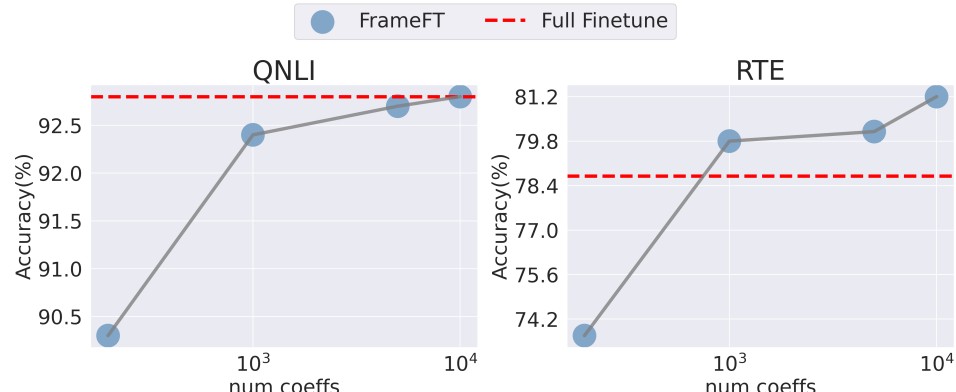

Figure 6: Performance of RoBERTa base model fine-tuned with FrameFT versus the number of non-zero coefficients on the GLUE benchmark. The red line indicates the performance of full fine-tuning.

## D    PERFORMANCE VERSUS SUBSPACE DIMENSION

In this experiment, we measure the performance of FrameFT for different values of subspace dimension, $\rho$. For this purpose, we fine-tune the RoBERTa-base model on two of the tasks in the GLUE benchmark - CoLA and MRPC, for different values of $\rho$. We keep the hyperparameters that we obtain for $\rho = 2$ as shown in Table 5. Figure 5 shows the performance of FrameFT as the subspace dimension is varied from 2 to 256. We observe that FrameFT performs well even with a subspace dimension of 2. We do not see a strong relationship between the performance and the subspace dimension. Given that the two key choices we have for adjustments – the number of coefficients and the subspace dimension – to fill the space, the number of coefficients is a more consistent way of improving performance as indicated in Figure 6. An alternative and promising direction for future work is to make the subspace dimension a trainable parameter, potentially enabling more adaptive and efficient optimization.

## E    PERFORMANCE AS A FUNCTION OF NUMBER OF NON-ZERO COEFFICIENTS

Figure 6 shows the performance of RoBERTa base fine-tuned with FrameFT on two of the tasks in the GLUE benchmark, in addition to the tasks shown in the main text. As noted earlier, we observe that FrameFT improves as the number of non-zero coefficients are increased, performing on par or better than full-finetuning.

| Optimizer | AdamW |
|---|---|
| LR scheduler | Linear schedule with warmup |
| Batch size | 32 |
| Head learning rate | 3E-3 |
| Adapter learning rate | 0.12 |
| Warmup ratio | 0.06 |
| Max. Seq. length | 512 |
| $\alpha$ | 20.0 |

Table 5: Hyperparameters for the GLUE benchmark

| Optimizer | AdamW |
|---|---|
| LR scheduler | Linear schedule with warmup |
| Batch size | 50 |
| Head learning rate | 3E-2 |
| Adapter learning rate | 0.33 |
| Warmup ratio | 0.06 |
| $\alpha$ | 250.0 |

Table 6: Hyperparameters for the Image classification task

| Optimizer | AdamW |
|---|---|
| LR scheduler | Linear schedule with warmup |
| Batch size | 128 |
| Learning rate | 0.1 |
| Warmup ratio | 0.03 |
| $\alpha$ | 200.0 |

Table 7: Hyperparameters for the Instruction tuning task

## F    HYPERPARAMETERS USED FOR THE EXPERIMENTS

We present the hyperparameters used for different experiments in Tables 5, 6, 7. As for the baselines, we use the hyperparameters suggested by the respective authors.

## G    FUSION FRAMES CONSTRUCTION: AN EXAMPLE

In this section, we briefly outline an algorithm for generating Tight Fusion Frames with the help of an example. Casazza et al. (2011) formalized a systematic framework for identifying the $(k, \rho, d)$ values for which a Tight Fusion Frame exists and generating the TFF whenever it exists. Their algorithm generates a TFF in $\mathbb{C}^d$. Since we are mainly interested in real vector spaces, we use the simple extension in Fickus et al. (2023b) to adapt the TFF to the real domain. The overall algorithm can be divided into three parts.

1. Construct a UNTF for $\mathbb{C}^\rho$ with $d$ elements by playing Spectral Tetris.
2. Modulate these vectors with $k^{\text{th}}$ roots of unity to form $k$ subspaces of dimension $\rho$ in $\mathbb{C}^d$.
3. Use the method in Fickus et al. (2023b) to extend the result to real-valued spaces.

We describe these steps in detail by walking through the construction of a (6,3,11) TFF, a Tight Fusion Frame spanning $\mathbb{C}^{11}$ with $k = 6$ subspaces where each subspace has a dimension of $\rho = 3$.

### G.1    SPECTRAL TETRIS

The first step is to generate a "smaller" frame and in the next step, we modulate the smaller frame to generate a "larger" Tight Fusion Frame. After generating a TFF for $\mathbb{C}^d$ we can easily extend it to the Real Field by applying the entrywise map $x + iy \mapsto \begin{bmatrix} x & -y \\ y & x \end{bmatrix}$. So, $k = 6, \rho = 3, d = 11$. As

the name suggests UNTFs are Tight frames where each frame vector has a unit norm. We construct a $4 \times 11$ matrix $F$ whose columns are the frame vectors for $\mathbb{C}^4$ which satisfies

In this first step, we generate a "smaller" frame - a Unit Norm Tight Frame (UNTF) for $\mathbb{C}^3$ with 11 vectors. We arrange these vectors in the columns of a matrix $F$. This UNTF is characterized by

- Columns with unit norm
- Rows are Orthogonal and have a constant norm, that is $FF^*$ is a constant multiple of the Identity matrix ( here the constant being $\frac{11}{3}$ )

We start by filling the first two entries in $F$ with 1

$$F = \begin{bmatrix} 1 & 1 & ? & ? & ? & ? & ? & ? & ? & ? & ? \\ ? & ? & ? & ? & ? & ? & ? & ? & ? & ? & ? \\ ? & ? & ? & ? & ? & ? & ? & ? & ? & ? & ? \end{bmatrix}$$

The remaining norm left to be filled is $\frac{11}{3} - 2 = \frac{5}{3}$. We continue to fill in 1s until the required norm is less than 1. Here, we can do this only once, yielding

$$F = \begin{bmatrix} 1 & 1 & 1 & ? & ? & ? & ? & ? & ? & ? & ? \\ ? & ? & ? & ? & ? & ? & ? & ? & ? & ? & ? \\ ? & ? & ? & ? & ? & ? & ? & ? & ? & ? & ? \end{bmatrix}$$

This leaves a norm of $\frac{2}{3}$ to be filled. This can be added with a $2 \times 2$ matrix $T(x)$. $T(x)$ here is defined as follows:

$$T(x) := \frac{1}{\sqrt{2}} \begin{bmatrix} \sqrt{x} & \sqrt{x} \\ \sqrt{2-x} & -\sqrt{2-x} \end{bmatrix}, \qquad T(x)T^*(x) = \begin{bmatrix} x & 0 \\ 0 & 2-x \end{bmatrix}$$

After substituting $T(x)$ with $x = \frac{2}{3}$, $F$ is now

$$F = \begin{bmatrix} 1 & 1 & 1 & \frac{1}{\sqrt{3}} & \frac{1}{\sqrt{3}} & 0 & 0 & 0 & 0 & 0 & 0 \\ 0 & 0 & 0 & \frac{\sqrt{2}}{\sqrt{3}} & -\frac{\sqrt{2}}{\sqrt{3}} & ? & ? & ? & ? & ? & ? \\ 0 & 0 & 0 & 0 & ? & ? & ? & ? & ? & ? & ? \end{bmatrix}$$

Now, we continue adding ones in row two until the norm becomes less than 1 again.

$$F = \begin{bmatrix} 1 & 1 & 1 & \frac{1}{\sqrt{3}} & \frac{1}{\sqrt{3}} & 0 & 0 & 0 & 0 & 0 & 0 \\ 0 & 0 & 0 & \frac{\sqrt{2}}{\sqrt{3}} & -\frac{\sqrt{2}}{\sqrt{3}} & 1 & 1 & ? & ? & ? & ? \\ 0 & 0 & 0 & 0 & ? & ? & ? & ? & ? & ? & ? \end{bmatrix}$$

Now we insert $T(x)$ with the remaining norm. We repeat this process until all the rows are filled. The Final $F$ is given by

$$F = \begin{bmatrix} 1 & 1 & 1 & \frac{1}{\sqrt{3}} & \frac{1}{\sqrt{3}} & 0 & 0 & 0 & 0 & 0 & 0 \\ 0 & 0 & 0 & \frac{\sqrt{2}}{\sqrt{3}} & -\frac{\sqrt{2}}{\sqrt{3}} & 1 & 1 & \frac{1}{\sqrt{6}} & \frac{1}{\sqrt{6}} & 0 & 0 \\ 0 & 0 & 0 & 0 & 0 & 0 & 0 & \frac{5}{\sqrt{6}} & -\frac{5}{\sqrt{6}} & 1 & 1 \end{bmatrix}$$

## G.2 MODULATION

In the second step of TFF construction, the $F$ matrix is modulated with complex roots of unity, one subspace at a time. For each $k_i = 0, 1, 2, \ldots k - 1$, we construct a row vector

$$w_{k_i} = \left[ \left( e^{\frac{i2\pi k_i}{k}} \right)^0 \left( e^{\frac{i2\pi k_i}{k}} \right)^1 \left( e^{\frac{i2\pi k_i}{k}} \right)^2 \ldots \left( e^{\frac{i2\pi k_i}{k}} \right)^{d-1} \right]$$

Each row of $F$ is multiplied by $w_{k_i}$ to produce the orthogonal basis for the subspace indexed by $k_i$. Theorem 14 by Casazza et al. (2011) proves that the Fusion Frames generated by this algorithm are Tight. The Final Fusion Frame generated is shown in Table 8.

$$
\begin{bmatrix}
1 & 1 & 1 & \frac{1}{\sqrt{3}} & \frac{1}{\sqrt{3}} & 0 & 0 & 0 & 0 & 0 & 0 \\
1 & \omega & \omega^2 & \frac{\sqrt{1}}{\sqrt{3}}\omega^3 & \frac{\sqrt{1}}{\sqrt{3}}\omega^4 & 0 & 0 & 0 & 0 & 0 & 0 \\
1 & \omega^2 & \omega^4 & \frac{\sqrt{1}}{\sqrt{3}}\omega^3 & \frac{\sqrt{1}}{\sqrt{3}}\omega^2 & 0 & 0 & 0 & 0 & 0 & 0 \\
1 & \omega^3 & 1 & \frac{\sqrt{1}}{\sqrt{3}}\omega^3 & \frac{\sqrt{1}}{\sqrt{3}} & 0 & 0 & 0 & 0 & 0 & 0 \\
1 & \omega^4 & \omega^2 & \frac{\sqrt{1}}{\sqrt{3}} & \frac{\sqrt{1}}{\sqrt{3}}\omega^4 & 0 & 0 & 0 & 0 & 0 & 0 \\
1 & \omega^5 & \omega^4 & \frac{\sqrt{1}}{\sqrt{3}}\omega^3 & \frac{\sqrt{1}}{\sqrt{3}}\omega^2 & 0 & 0 & 0 & 0 & 0 & 0 \\
0 & 0 & 0 & \frac{\sqrt{2}}{\sqrt{3}} & -\frac{\sqrt{2}}{\sqrt{3}} & 1 & 1 & \frac{1}{\sqrt{6}} & \frac{1}{\sqrt{6}} & 0 & 0 \\
0 & 0 & 0 & \frac{\sqrt{2}}{\sqrt{3}}\omega^3 & -\frac{\sqrt{2}}{\sqrt{3}}\omega^4 & \omega^5 & 1 & \frac{1}{\sqrt{6}}\omega & \frac{1}{\sqrt{6}}\omega^2 & 0 & 0 \\
0 & 0 & 0 & \frac{\sqrt{2}}{\sqrt{3}} & -\frac{\sqrt{2}}{\sqrt{3}}\omega^2 & \omega^4 & 1 & \frac{1}{\sqrt{6}}\omega^2 & \frac{1}{\sqrt{6}}\omega^4 & 0 & 0 \\
0 & 0 & 0 & \frac{\sqrt{2}}{\sqrt{3}}\omega^3 & -\frac{\sqrt{2}}{\sqrt{3}} & \omega^3 & 1 & \frac{1}{\sqrt{6}}\omega^3 & \frac{1}{\sqrt{6}} & 0 & 0 \\
0 & 0 & 0 & \frac{\sqrt{2}}{\sqrt{3}} & -\frac{\sqrt{2}}{\sqrt{3}}\omega^4 & \omega^2 & 1 & \frac{1}{\sqrt{6}}\omega^4 & \frac{1}{\sqrt{6}}\omega^2 & 0 & 0 \\
0 & 0 & 0 & \frac{\sqrt{2}}{\sqrt{3}}\omega^3 & -\frac{\sqrt{2}}{\sqrt{3}}\omega^2 & \omega^1 & 1 & \frac{1}{\sqrt{6}}\omega^5 & \frac{1}{\sqrt{6}}\omega^4 & 0 & 0 \\
0 & 0 & 0 & 0 & 0 & 0 & 0 & \frac{5}{\sqrt{6}} & -\frac{5}{\sqrt{6}} & 1 & 1 \\
0 & 0 & 0 & 0 & 0 & 0 & 0 & \frac{5}{\sqrt{6}}\omega & -\frac{5}{\sqrt{6}}\omega^2 & \omega^3 & \omega^4 \\
0 & 0 & 0 & 0 & 0 & 0 & 0 & \frac{5}{\sqrt{6}}\omega^2 & -\frac{5}{\sqrt{6}}\omega^4 & 1 & \omega^2 \\
0 & 0 & 0 & 0 & 0 & 0 & 0 & \frac{5}{\sqrt{6}}\omega^3 & -\frac{5}{\sqrt{6}} & \omega^3 & 1 \\
0 & 0 & 0 & 0 & 0 & 0 & 0 & \frac{5}{\sqrt{6}}\omega^4 & -\frac{5}{\sqrt{6}}\omega^2 & 1 & \omega^4 \\
0 & 0 & 0 & 0 & 0 & 0 & 0 & \frac{5}{\sqrt{6}}\omega^5 & -\frac{5}{\sqrt{6}}\omega^4 & \omega^3 & \omega^2
\end{bmatrix}
$$

Table 8: $(\mathbf{6}, \mathbf{3}, \mathbf{11})$-**TFF** for $\mathbb{C}^{11}$. Here, $\omega = e^{i\pi/3}$. Each pair of rows belongs to the same subspace if their indices differ by a multiple of 6

