# OpenReview forum: "Fine-Tuning of Transformer models with Frames"
_ICLR.cc/2026/Conference — Submitted to ICLR 2026_

### Official Review · Reviewer_6i8s · 2025-10-28

**Soundness:** 2
**Presentation:** 2
**Contribution:** 2
**Rating:** 2
**Confidence:** 5

**Summary:**

This paper introduces a new parameter efficient fine tuning (PEFT) method that seeks to model the adapter weights in a Fusion Frame representation space. More precisely the authors show that by modelling the adapter wights as a sparse matrix that lives in a fusion frame representation space the memory footprint and parameter count can be reduced leading to a more efficient PEFT methodology. In order to show that their method yields good performance the authors showcase their method on a collection of supervised fine tuning benchmarks associated to language tasks as well apply their method to some vision tasks.

**Strengths:**

**Originality:** The idea of using sparse adaptors in a fusion frame representation space as adaptors for fine tuning is original as at least from my knowledge of PEFT methods that are currently used within the literature. However, I don't feel the main theorem, namely theorem 3.1 is very original as it uses lemma 1 which is quite a simple lemma that follows essentially from the definition of a fusion frame. While such theorems are not always seen in papers on PEFT methods they are prevalent in many deep learning paper on optimization. Furthermore, the authors don't actually show Theorem 3.1 for a deep network. This seems to be stated for one layer, unless I have miss read something.

**Clarity:** The paper is written well and was easy for me to follow. In general the authors did a good job of clarifying their methods and their ideas and explaining the results of their experiments.

**Weaknesses:**

**Novelty:** It is my feel that the paper lacks novelty. While their idea of looking at a PEFT adaptors from the point of view of fusion frames is nice and new I can't say that there is anything deep and novel going on. I even found the experimental performances on the language tasks shown in the main paper rather underwhelming. If I compare their approach to FourierFT for each model I found that for 3 out of the 5 models it achieves comparable performance to FourierFT and for Gemma-2-2B and Llama-3.1-8b it achieves better performance but only by about approximately 1.5% on the average. I also found this for the result on the GLUE benchmark shown in Table 1, although in the case of the RoBERTa Base their approach beats FourierFT by 1.1%. Furthermore, I didn't find anything novel in the theory. In Theorem 3.1 you are basically showing that a function that has been adapted using fusion frames can converge linearly under gradient descent. But this is for a function not a deep model. In the case of a deep model you have such fusion frame adaptors in each layer so you would need to model a composition. This means you Theorem 3.1 doesn't really apply. Also, I noticed you didn't have any limitations of your methodology.

**Significance:** I don't think the paper will make a significant impact on the community as it seems to just show how one can take a certain different representation of adaptors with no real theoretical insight into why this works out well. Even from the experiments it does comparable to FourierFT in many cases. The results on the ViT's in the appendix seemed to show that their approach did quite well when compared to the others though the authors only considered a ViT-B and a ViT-L model which themselves are rather outdated.

**Questions:**

1. How is Theorem 3.1 useful in the case that my function is an LLM with adaptors employing your fusion frame approach? I think here what you have proved is useful for a one layer model but you need another theorem that shows how the bound changes when you compose functions as deep models are compositions of the layers.

2. I am guessing your point of Theorem 3.1 is to show that your adaptation method leads to a bound on the gradient that can be used to show the rate of convergence of gradient descent? Did you find that with fusion frame adaptation converged faster than the other methods you tested against in the experiments section. I noticed that the scores when compared to say FourierFT were comparable and so my hypothesis would be that your adaptation method would only lead to marginal improvements in convergence speed. Please clarify this for me. Furthermore, don't you fine tune your models with the Adam optimizer? So is Theorem 3.1 really relevant?

3. Do you apply your fusion frame adaptation method to both the attention matrices and feedforward matrices or just one out of the 2? I could have missed this but if this is explained in the paper could you point me towards where you explain how exactly you apply your method in the actual transformer architecture for the LLM experiments and the ViT experiments in the appendix.

4. I noticed you did not really talk about any limitations of your method or at least position some limitations with respect to what is currently done in the literature. Are there any limitations to your method?

---

> ### Author Response · Authors · 2025-11-21
> **Reply by authors**
>
> We thank the reviewer for their time and effort in reviewing our work. Please find our responses to the questions below.
>
> > W.1 If I compare their approach to FourierFT for each model I found that for 3 out of the 5 models it achieves comparable performance to FourierFT and for Gemma-2-2B and Llama-3.1-8b it achieves better performance but only by about approximately 1.5% on the average. I also found this for the result on the GLUE benchmark shown in Table 1, although in the case of the RoBERTa Base their approach beats FourierFT by 1.1%.
>
> We agree with the reviewer's observation but we believe it reflects our expectation. Any well-studied orthonormal basis—whether Fourier, Wavelet, or Frames should deliver roughly comparable downstream accuracy when used for parameter updates unless it is strictly less expressive (e.g., rank deficient). It would be surprising if one basis consistently dominated others across diverse models and tasks without incorporating some domain-specific structure. The fact that FrameFT and FourierFT achieve similar accuracy is not a weakness. It confirms that both provide sufficiently expressive parameterizations.
>
> The critical point to note is _how_ these comparable accuracies are achieved. Consider the full picture from Table 3: FrameFT matches FourierFT's performance while using identical parameter counts. This is OK. However, frameFT achieves 20x higher inference throughput (Table 2). This is not a marginal difference. FrameFT exceeds even LoRA's throughput. The performance gap comes from structural differences: FourierFT requires dense matrix operations in the Fourier domain, while we exploit the inherent block-sparse structure in Fusion Frames.
>
> Beyond throughput, FrameFT offers advantages in multi-adapter deployment scenarios. We require only 1.28MB per task-specific adapter because fusion frames can be generated algorithmically and shared across adapters. For systems serving dozens of task-specific models, this gives a qualitative difference in feasibility. In light of this use case, we believe that the relevant question for PEFT methods is not which method yields 1% higher accuracy. Instead, which achieves comparable accuracy with superior efficiency? We hope that with this clarification, you will agree that the comparison is decisive.

---

> > ### Author Response · Authors · 2025-11-21
> > **Author reply**
> >
> > > W.2  In Theorem 3.1 you are basically showing that a function that has been adapted using fusion frames can converge linearly under gradient descent. But this is for a function not a deep model. In the case of a deep model you have such fusion frame adaptors in each layer so you would need to model a composition. This means you Theorem 3.1 doesn't really apply.
> >
> > We realise this point may have been unclear and likely affected your final decision. We have outlined the context below to help clear up the confusion.
> >
> > The function $f$ in Theorem 3.1 is the loss function of the complete network, not a single-layer function. For a network with $L$ layers and parameters $W=(W_1, \cdots, W_L)$, the loss function is $f(W)=\mathcal{L}(\text{Model}(x;W),y)$. This loss function already encodes the composition of all layers. When we analyse convergence, we analyse $f(W)$ as a function from the parameter space to $\mathbb{R}$, treating the layer composition as implicit in $f$. FrameFT reparameterizes each weight matrix as $W_l = W_{l,0} + P_{m,l}C_l P_{n,l}$. The loss then becomes $f(C)=f(C_1,\cdots,C_L)$ where $C=(C_1, \cdots, C_L)$ gives all coefficient matrices. Lemma 1 establishes that if $f(W)$ is L-Lipschitz smooth in the weight space, then $f(C)$ is also smooth in the coefficient space.
> >
> >
> > This is standard and exactly how Sun (2024) analyses LoRA convergence properties for multi-layer networks where the loss is a function of all LoRA parameters. Similar with
> > Malinovsky (2024), where they look at LoRA chains.
> >
> > The alternative formulation you suggest, where we analyse each layer's output as a function of its parameters, will involve Jacobians, significantly complicating analysis without giving additional insight for convergence guarantees.
> >
> > ------
> > Sun (2024) Youbang Sun, Zitao Li, Yaliang Li, and Bolin Ding. Improving loRA in privacy-preserving federated learning, 2024.
> >
> > Malinovsky (2024) Grigory Malinovsky, Umberto Michieli, and others, 2024. Randomized asymmetric chain of LoRA: The first meaningful theoretical framework for low-rank adaptation.

---

> > > ### Author Response · Authors · 2025-11-21
> > > **Author reply**
> > >
> > > > Q.1 How is Theorem 3.1 useful in the case that my function is an LLM with adaptors employing your fusion frame approach? I think here what you have proved is useful for a one layer model but you need another theorem that shows how the bound changes when you compose functions as deep models are compositions of the layers.
> > >
> > > The _function_ here is the _loss function_ of the deep model, which includes _all_ the layers in the model. This is a standard technique used in related convergence analysis works, e.g., in REFS Sun (2024) and Malinovsky (2024) above.
> > >
> > > > Q.2 I am guessing your point of Theorem 3.1 is to show that your adaptation method leads to a bound on the gradient that can be used to show the rate of convergence of gradient descent? Did you find that with fusion frame adaptation converged faster than the other methods you tested against in the experiments section. I noticed that the scores when compared to say FourierFT were comparable and so my hypothesis would be that your adaptation method would only lead to marginal improvements in convergence speed. Please clarify this for me. Furthermore, don't you fine tune your models with the Adam optimizer? So is Theorem 3.1 really relevant?
> > >
> > > The main question here is how the theory informs practice. Your intuition is partially correct but misses some important aspects of our analysis.
> > >
> > > The purpose of our analysis is not to claim faster convergence than other methods, but rather to provide design guidance through the bound $\tilde{L} = L\sqrt{r} B_m B_n$. This bound tells us that the Lipschitz constant and hence the optimization landscape can nicely depend on two design choices we have control over: the frame bounds ($B_m, B_n$) and the redundancy. This suggests setting $B_m = B_n = 1$ with small redundancy ($k\rho = d$), which is precisely what we do.
> > >
> > > Yes, you correctly observed that FrameFT and FourierFT achieve comparable accuracy, which we discussed in our response above. Our analysis generalizes to any basis transformation method where one can replace $\mathbf{P}_m, \mathbf{P}_n$ with Fourier bases and this will give results for FourierFT, with SVD components will give SVFit, etc. Well-structured bases (Fourier, Wavelets, etc.) can typically be normalized to achieve similar frame bounds. However, while these methods may have similar optimization properties, they can differ dramatically in compute efficiency. Our fusion frames achieve both favorable frame bounds and inherent block-sparse structure, yielding strong throughput advantage.
> > >
> > > Finally, regarding the comment about gradient descent versus Adam. Is GD analysis irrelevant if we train with Adam? We know that adaptive methods like Adam build on gradient descent principles: they indeed use the same gradients but with adaptive step sizes and momentum. Lipschitz smoothness guarantees for GD directly inform learning rate selection and convergence behavior for Adam. Lemma 1 shows that FrameFT's reparameterization preserves smoothness of the loss landscape. This is a structural property that benefits any gradient-based optimizer, not just GD alone. Separately, you will appreciate that Sun (2024) argues that LoRA's reparameterization creates non-smooth landscapes, which causes instability regardless of whether one uses GD, SGD, or Adam. So, our guarantee of smoothness preservation is optimizer-agnostic and practically valuable.
> > >
> > > > Q.3 Do you apply your fusion frame adaptation method to both the attention matrices and feedforward matrices or just one out of the 2? I could have missed this but if this is explained in the paper could you point me towards where you explain how exactly you apply your method in the actual transformer architecture for the LLM experiments and the ViT experiments in the appendix.
> > >
> > > We follow the LoRA recipe of updating the Query and Value matrices of each transformer block. We describe the experimental setup within the subsection for each experiment. Here are the line numbers where we describe the experimental setup
> > > - Lines 301-305 for the GLUE benchmark experiments
> > > - Lines 327-331 for the Instruction tuning experiments
> > > - Lines 898-910 for the Vision Transformer experiments

---

> > > > ### Author Response · Authors · 2025-11-21
> > > > **Author reply**
> > > >
> > > > > Q.4 I noticed you did not really talk about any limitations of your method or at least position some limitations with respect to what is currently done in the literature. Are there any limitations to your method?
> > > >
> > > > We briefly note the lack of efficient kernels optimized for different hardware accelerators in the conclusion of the paper. Here are some limitations of our work and possible avenues for improvement,
> > > > 1. We have only evaluated FrameFT on open-weight models of up to 13B parameters to keep the computational costs of our experiments reasonable. While the general idea should work for much larger models, the performance profile has not yet been benchmarked for these models.
> > > > 2. As briefly noted in the paper, while sparsity is an important feature of our formulation (frames, coefficients), and offers benefits, thoroughly exploiting it for efficiency gains will need more work (such as developing specialized kernels for various hardware accelerators).
> > > >
> > > > We hope we answered all the questions that were raised and provided a better understanding of the benefits of our work and how it differentiates FrameFT from the current PEFT methods. We are available to answer any further questions you may have.

---

### Official Review · Reviewer_AUxt · 2025-10-28

**Soundness:** 2
**Presentation:** 3
**Contribution:** 3
**Rating:** 4
**Confidence:** 5

**Summary:**

This paper proposes an efficient parameter fine-tuning method based on Fusion Frames, FrameFT, for efficiently adapting large-scale Transformer models in language and visual tasks. This method significantly reduces trainable parameters and memory usage while maintaining or even improving performance by representing parameter updates with sparse coefficients in multiple overlapping subspaces. The main contribution of this paper lies in introducing the Frame theory into the model fine-tuning framework, providing proofs of convergence and smoothness, and verifying the universality of the method in instruction fine-tuning, language and visual tasks.

**Strengths:**

1. The FrameFT parameters and memory efficiency are significant. The weight update is expressed by the sparse coefficients of the overlapping subspace, reducing the trainable parameters and video memory occupation, while maintaining or improving performance.
2. It has sufficient cross-modal verification, performs excellently in both language and visual tasks, and has strong universality.

**Weaknesses:**

1. The theoretical analysis of the method in the paper is rather abstract. Although there is a formal proof, it lacks an intuitive mechanism explanation for the FrameFT structure to enhance optimization and generalization performance.
2. The cross-layer shared sparse model lacks empirical support. It is assumed that this structure can be shared across layers, but sufficient verification has not been provided.

**Questions:**

The author can further provide a theoretical explanation or mathematical analysis of the "adaptive sparse strategy" to clarify the core logic of designing sparse positions based on task loss and gradient sensitivity, as well as the theoretical advantages of this strategy over random sparsity. Meanwhile, it is suggested that the performance improvement effect of the adaptive strategy be demonstrated through comparative experiments of complex tasks (such as the detailed classification of FGVC Aircraft) and the visualization of the dynamic adjustment process of sparse positions, so as to further deepen the understanding of the optimization direction of sparse patterns.

---

> ### Author Response · Authors · 2025-11-21
> **Author reply**
>
> We thank the reviewer for their time and effort in reviewing our work. Here are our responses to the questions raised.
> > W.1 The theoretical analysis of the method in the paper is rather abstract. Although there is a formal proof, it lacks an intuitive mechanism explanation for the FrameFT structure to enhance optimization and generalization performance.
>
> First, we provide some general intuition about FrameFT vis-à-vis optimisation, and explain the underlying strategy of the proof.
>
> **Intuition 1)** Lemma 1 shows that FrameFT preserves Lipschitz smoothness of the loss. Sun (2024) shows that LoRA does not. From a practical side, we know that a smooth loss landscape means that small parameter changes result in small changes in loss. Gradients are stable. Learning rates are reliable. LoRA's non-smooth landscape can cause training instability. Some recent works seek to speed up LoRA using refined initialisation - Li (2024) or learning rate ideas - Hayou (2024). This can be viewed as encouraging stability. The asymmetric behaviour between LoRA's $A$ and $B$ matrices in Malinovsky (2024) and Hayou (2024) comes from this non-smoothness. FrameFT's fusion frame structure avoids these pathologies by change of basis.
>
> **Intuition 2)** Our bound suggests that optimisation difficulty depends on frame bounds and redundancy, suggesting concrete steps we can take: (a) Use Parseval frames $B_m=B_n=1$
> to keep the smoothness constant small, which preserves signal energy without amplification, keeping gradients well-behaved. (b) Use small redundancy. These ideas come directly from understanding the bound, not trial and error. These results were unavailable in the literature to our knowledge.
>
> **Intuition 3)** The fusion frame structure regularises adaptation by forcing updates to be expressible as sparse combinations of geometrically well-separated subspaces (e.g., equichordal). Our empirical results show that this does not hurt performance while reducing parameters and improving efficiency, suggesting it is a sensible inductive bias.
>
> Finally, let us also clarify the main points that make smoothness preservation work:
>
>   The proof (Appendix A) hinges on how gradients change under our reparameterization. For LoRA, the gradient with respect to $ (A, B) $ involves products like $\nabla_B f = (\nabla_W f) A^T$ and $\nabla_A f = B^T (\nabla_W f)$, creating an asymmetric coupling where changes in $A$ affect $B$'s gradient and vice versa. The analysis in Sun (2024) and Hayou (2024) suggests that this is clearly undesirable.
>
> For FrameFT, on the other hand, the gradient transformation is $\nabla_{C_l} f = P_{l}^T (\nabla_{W_l} f) P_{prev}$. It is a coordinate change via orthonormal projections, much like whitening transformations in statistics or a change of basis in linear algebra. We are viewing the same gradient information in a different coordinate system. Assuming Parseval's Frame for simplicity, it is not hard to show that
>
> $ \lvert \lvert \nabla_{C_l} f(C^1) - \nabla_{C_l} f(C^2)\  \rvert \rvert_F \leq \lvert \lvert \nabla_{W_l} f(W^1) - \nabla_{W_l} f(W^2) \rvert \rvert_F $
>
> Where the right-hand term is bounded because of Lipschitz smoothness. This enables the fusion frame transformation to preserve the gradient Lipschitz constant rather than amplifying it, and so FrameFT maintains smooth optimisation landscapes.
>
> ------
> Sun (2024) Youbang Sun, Zitao Li, Yaliang Li, and Bolin Ding. Improving loRA in privacy-preserving federated learning, 2024.
>
> Li (2024) Bingcong Li, Liang Zhang, Aryan Mokhtari, and Niao He. On the crucial role of initialization for matrix
> factorization.
>
> Hayou (2024) Soufiane Hayou, Nikhil Ghosh, and Bin Yu. Lora+: Efficient low rank adaptation of large
> models
>
> Malinovsky (2024) Grigory Malinovsky, Umberto Michieli, and others, 2024. Randomized asymmetric chain of LoRA: The first meaningful theoretical framework for low-rank adaptation.

---

> ### Author Response · Authors · 2025-11-21
> **Author reply**
>
> > W.2 The cross-layer shared sparse model lacks empirical support. It is assumed that this structure can be shared across layers, but sufficient verification has not been provided.
>
> The sparsity pattern determines which subspace interactions are active. Our performance across benchmarks shows that shared sparsity patterns do not limit expressiveness. Each layer may differ in the strength of these interactions captured by coefficient values, not in which interactions exist (captured by shared sparsity pattern). This suggests that important subspace interactions are relatively common across layers. Other methods, such as FourierFT and SVFit, also share their sparsity pattern across layers. To stay competitive, with a similar parameter count, using a shared pattern across layers is necessary.
>
> FrameFT shares the sparsity pattern (which subspace pairs interact) while keeping the interaction strengths (coefficient values) layer-specific and continuously optimized. Even with this constraint, we demonstrate that FrameFT performs at par or better than the baseline methods on different benchmarks. If cross-layer sharing severely limited expressiveness, we would expect to see systematic performance degradation. In practice, we can relax this constraint and allow a more flexible sparsity pattern across layers; each layer can be tuned independently to further improve the performance of our method.
>
> > Q. The author can further provide a theoretical explanation or mathematical analysis of the "adaptive sparse strategy" to clarify the core logic of designing sparse positions based on task loss and gradient sensitivity, as well as the theoretical advantages of this strategy over random sparsity. Meanwhile, it is suggested that the performance improvement effect of the adaptive strategy be demonstrated through comparative experiments of complex tasks (such as the detailed classification of FGVC Aircraft) and the visualization of the dynamic adjustment process of sparse positions, so as to further deepen the understanding of the optimization direction of sparse patterns.
>
> We are sorry, but we do not understand this question. We do not use an adaptive sparse strategy scheme. From the question, it appears that your interpretation is that we use task loss or gradient sensitivity to figure out the sparsity pattern. We do not use these strategies as well. Our sparsity pattern within the blocks is random. We are happy to provide additional clarifications if you can point out a specific section in the paper.

---

> > ### Author Response · Authors · 2025-11-21
> > **Reply by authors**
> >
> > Through the rebuttal, we believe we have fully addressed all the points raised and are happy to provide any further clarification that may be needed! We hope this strengthens your support of our work.

---

### Official Review · Reviewer_Z3X9 · 2025-10-29

**Soundness:** 3
**Presentation:** 3
**Contribution:** 2
**Rating:** 4
**Confidence:** 4

**Summary:**

This paper introduces FrameFT, a new parameter-efficient fine-tuning method based on Fusion Frames. Fusion Frames are pre-computed by decomposing the input and output spaces into multiple subspaces. Interactions between corresponding input–output subspace pairs are captured using a learnable sparse matrix. The Fusion Frames and sparsity pattern are shared across layers, which provides computational benefits. Experiments on both language and vision tasks demonstrate the effectiveness of FrameFT.

**Strengths:**

- The paper is generally clear and well structured.

- The proposed FrameFT method is a novel application of Fusion Frame theory to PEFT. The experimental evaluation spans both NLP and vision domains, providing supporting evidence of its effectiveness.

- Theoretical analysis is provided regarding convergence and computational efficiency, which strengthens the technical contribution.

**Weaknesses:**

- The proposed method appears closely related to existing PEFT techniques, particularly SVFT [1], which also decomposes model weights and introduces a learnable sparse interaction matrix. SVFT additionally explores off-diagonal parameters to capture richer basis interactions. FrameFT instead uses pre-computed Tight Fusion Frames (TFFs) to partition the input and output spaces into fixed-dimension subspaces, but similarly employs a layer-shared sparse interaction matrix. Given these conceptual overlaps, the novelty of FrameFT would benefit from a more explicit discussion of the differences, both theoretically and empirically.

- A direct comparison with recent and relevant PEFT baselines such as SVFT [1], SMT [2], and VeRA [3] is currently missing. Such comparisons are important to assess whether FrameFT provides meaningful gains over the state-of-the-art.

---

References

[1] Lingam et al., SVFT: Parameter-Efficient Fine-Tuning with Singular Vectors, NeurIPS 2024

[2] He et al., SMT: Fine-Tuning Large Language Models with Sparse Matrices, ICLR 2025

[3] Kopiczko et al., VeRA: Vector-Based Random Matrix Adaptation, ICLR 2024

**Questions:**

1. Lines 368–369 mention that constructing Fusion Frames requires only a finite amount of time. Could the authors provide the actual wall-clock time (in seconds) for the configurations used in the reported experiments?

2. The abstract (lines 18–20) states that Fusion Frames can be shared across layers. Were these frames computed once and reused across all compatible layers, or recomputed per layer? Clarification would help better understand the computational trade-offs.

---

> ### Author Response · Authors · 2025-11-21
> **Author reply**
>
> We thank the reviewer for their time and effort in reviewing our work. Please find our responses to the questions below.
>
>
> > W.1 The proposed method appears closely related to existing PEFT techniques, particularly SVFT [1], which also decomposes model weights and introduces a learnable sparse interaction matrix. SVFT additionally explores off-diagonal parameters to capture richer basis interactions. FrameFT instead uses pre-computed Tight Fusion Frames (TFFs) to partition the input and output spaces into fixed-dimension subspaces, but similarly employs a layer-shared sparse interaction matrix. Given these conceptual overlaps, the novelty of FrameFT would benefit from a more explicit discussion of the differences, both theoretically and empirically.
>
> While SVFT and FrameFT both use sparse coefficient matrices, they differ a fair bit in how the underlying basis is constructed, leading to very different properties and practical trade-offs.
>
> SVFT extracts singular vectors from pretrained weights. It is an excellent data-driven approach where the basis adapts to the specific initialization of each layer. FrameFT constructs bases using frame theory, where the basis is determined by geometric principles *independent* of pretrained weights. This distinction is important because SVFT's basis varies across layers and needs recalculation for layers or different models. The basis in FrameFT are universal for a fixed dimension and can be shared across models, layers, and even tasks.
>
> SVFT computes and must store singular vectors for each layer. For a parameter matrix $W$, this requires storing $U$ and $V$ per layer. For each parameter matrix $\mathbb{R}^{n \times n}$, this means two more $\mathbb{R}^{n \times n}$ matrices for the left and right singular vectors, making the memory footprint much larger.
> FrameFT generates dimension-appropriate fusion frames algorithmically just once and shares them across all layers of the same dimension, requiring zero additional storage. Finally, our frames have an inherent block-sparse structure by design (Table 8), giving the throughput advantages shown in Table 2.
>
> > W.2 A direct comparison with recent and relevant PEFT baselines such as SVFT [1], SMT [2], and VeRA [3] is currently missing. Such comparisons are important to assess whether FrameFT provides meaningful gains over the state-of-the-art.
>
> Thank you for suggesting these comparisons. Below we show the comparison of FrameFT with SMT, SVFT, and VeRA for finetuning RoBERTa Large model on the GLUE benchmark. As the table below suggests, FrameFT performs comparable/better than these baselines while using the fewest number of parameters.
>
> | Method  | **#params** | SST-2    | MRPC     | CoLA     | QNLI     | RTE      | STS-B    | Avg      |
> | :-----: | ----------- | -------- | -------- | -------- | -------- | -------- | -------- | -------- |
> |   SMT   | 1.5M        | 96.0     | 89.7     | 69.4     | 93.6     | 84.1     | 91.8     | 87.4     |
> |  $\text{SVFT}^R_{d=2}$   | 0.25M       | 96.1     | 90.2     | 66.7     | 94.3     | 83.0     | **92.1** | 87.1     |
> |  VeRA   | 61K         | 96.1     | 90.9     | 68.0     | **94.4** | 85.9     | 91.7     | 87.8     |
> | FrameFT | 48K         | **96.2** | **92.6** | **69.8** | 93.4     | **88.1** | 91.9     | **88.7** |
>
>
> These performance numbers, along with the parameter savings and throughput improvements, make FrameFT a compelling choice for PEFT. While the accuracy numbers are reassuring, the meaningful distinctions lie in efficiency, i.e., parameter count and inference throughput.

---

> > ### Author Response · Authors · 2025-11-21
> > **Author reply**
> >
> > > Q.1 Lines 368–369 mention that constructing Fusion Frames requires only a finite amount of time. Could the authors provide the actual wall-clock time (in seconds) for the configurations used in the reported experiments?
> >
> > Here is the time taken to generate the Fusion Frames for Llama-2, Gemma-2 and Llama-3.1-8B models
> >
> > |    Model     | **time(ms)** |
> > | :----------: | ------------ |
> > |  Llama-2-7B  | 89.8         |
> > | Llama-2-13B  | 110.9        |
> > |  Gemma-2-2B  | 117.0        |
> > |  Gemma-2-9B  | 211.6        |
> > | Llama-3.1-8B | 114.3        |
> >
> > Note that this only needs to be done once during initialization. In the scenario where a single backbone model is serving multiple adapters, the Fusion Frames remain completely unchanged; switching adapters requires only a hot-swap of the non-zero coefficients. Hence, the computational cost is effectively amortized throughout the service of the model.
> >
> > > Q.2 The abstract (lines 18–20) states that Fusion Frames can be shared across layers. Were these frames computed once and reused across all compatible layers, or recomputed per layer? Clarification would help better understand the computational trade-offs.
> >
> > Yes, we compute the Fusion Frames just once during the initialization and reuse them across all the layers. We kept the Fusion Frame parameters $(k, \rho, d)$ the same for all the layers, so the Fusion Frames can be shared.
> > Our current experiments use uniform settings for $(k,\rho,d)$ across all layers, which already achieves competitive performance. However, fusion frames support layer-specific subspace dimensions, for instance, using larger $\rho$ in early layers to capture richer representations. We believe this flexibility could be valuable in other structured adaptation settings like continual learning, where new tasks could reuse the fixed fusion frame structure while only updating sparse coefficients. In this way, task addition may be possible without catastrophic forgetting. Another use case is federated learning, where the frames could be shared globally while clients train only local sparse coefficients, reducing communication cost. These are promising directions.
> >
> > We hope that we have addressed all the questions that were raised and provided a better understanding of the benefits of our work and how it differentiates from the current PEFT methods. We are available to answer any further questions you may have.

---

### Official Review · Reviewer_TDeP · 2025-10-31

**Soundness:** 2
**Presentation:** 2
**Contribution:** 2
**Rating:** 4
**Confidence:** 4

**Summary:**

The paper proposes a new PEFT method comparable to LoRA for transformer models. Instead of LoRA's low-rank factorization, they adopt the multiple overlapped subspace projections that are motivated from the fusion frame theory. More specifically they select the input output subspace projection matrices from the spectral tetris algorithm (Casazza et al. 2011), fix them, and only learn the inner full coefficient matrices. The method was tested on several language benchmarks (also on some vision benchmarks in the appendix) while compared with some selected PEFT methods like Fourier-FT, S2FT, DoRA, and LoRA.

**Strengths:**

- Interesting that they used the less popular (to my knowledge) theory of fusion frame to the PEFT problem.
- In some scenarios/settings, they showed improvement in performance over some PEFT methods on various benchmarks.

**Weaknesses:**

- First, it would be nice to provide some reference/book on the frame theory in Sec.2 (background) unless you are proposing it in the paper. You can do it at the beginning of the section, like "The readers can refer to this citation for the introduction of the theory..."

- In Sec.3 (main) they proposed to replace the left right singular matrices in LoRA by the fusion frame matrices. It looks to me like a logical leap, and there is not much discussion on intuition and motivation. So, why frame theory? How does frame theory play a role here? The fusion frame matrices, assuming Parseval as you actually did, appear to be nothing but adopting overcomplete sets of orthonormal basis in place of the low-rank left and right singular vectors.

- The proposed method also looks similar to LoRA-XS (https://arxiv.org/pdf/2405.17604) which fixes the singular vectors and trains the full inner matrix only. But because in theory we can stack multiple modules or multiple heads in LoRA-XS, it can potentially be similar to having a set of overcomplete orthonormal bases in the end?

LoRA-XS: Low-Rank Adaptation with Extremely Small Number of Parameters Klaudia Bałazy, Mohammadreza Banaei, Karl Aberer, Jacek Tabor

- Also, is there any ablation study on how effective the chosen spectral tetris algorithm is? Ie, one can easily think of using randomly generated overcomplete orthonormal basis for Pn and Pm via multiple runs of Gram-Schmidt processes and stacking them. Why spectral tetris?

- Why is Theorem 3.1 (convergence analysis) especially unique for the proposed method? I guess it also holds for LoRA and any other variants (eg, LoRA-XS) if we make some bounded parameter assumptions since most PEFT models are Lipschitz smooth. In this sense, Line 161, where they said LoRA can fail to converge, may be an overstatement?

- Another aspect that is not well discussed in the paper is the memory footprint and the inference efficiency. LoRA's efficient memory use is one of its key benefits: LoRA can only store two low-rank matrices. But the proposed approach needs to store the large overcomplete basis matrices (I guess more than (d x d)) or merge into (d x d) matices, which can be less efficient than LoRA especially for on-device applications where many PEFT methods are targeted for. LoRA can use far less memory footprint during forward pass (run sequentially A@x then B@(A@x)), but the proposed approach uses the overcomplete basis (although fixed during training, but have to be stored as a part of the checkpoints, and loaded at the inference time). The number of trainable parameters may not be a big deal in practice, but the inference time matters more. I guess the proposed method has little benefit in this regard.

- I am a bit skeptical about the fact that there are too many similar papers these days on sparse PEFT methods that extend LoRA in a variety of different ways to structure the learnable parameter space, eg, Fourier, SVD, vector-based, partially fixing some parameters and varying the rest, etc. Since most ideas are essentially tweaking a bit the parameter space of the (already powerful) pre-trained model, and there is always a high chance that the fine-tuned model performs better than the pre-trained one. So it may be hard to judge which parameter structure is better than others in a uniform manner in a principled way. I mean showing the performance improvements on some selected scenarios (hyperparameter settings and chosen benchmarks) could be potentially possible for most reasonable parametrization. This is not to criticize your paper, but I think some rigorous theoretical study on the relationship between parameter space and model expressibility needs to be done.

**Questions:**

See questions in the weakness section.

---

> ### Author Response · Authors · 2025-11-21
> **Author reply**
>
> We thank the reviewer for their time and effort in reviewing our work. Please find our responses to the questions below.
>
> > W.1 First, it would be nice to provide some reference/book on the frame theory in Sec.2 (background) unless you are proposing it in the paper. You can do it at the beginning of the section, like "The readers can refer to this citation for the introduction of the theory..."
>
> We have references to textbooks when we first introduced Frame Theory, in line 75. In section 2, we wanted to summarize concepts from Frame Theory relevant to the paper, so, its easy for the reader to follow through. However, based on your suggestion, we will add new text at the beginning of Section 2 referencing text books that the reader can check.
>
> > W.2 In Sec.3 (main) they proposed to replace the left right singular matrices in LoRA by the fusion frame matrices. It looks to me like a logical leap, and there is not much discussion on intuition and motivation. So, why frame theory? How does frame theory play a role here? The fusion frame matrices, assuming Parseval as you actually did, appear to be nothing but adopting overcomplete sets of orthonormal basis in place of the low-rank left and right singular vectors.
>
> We appreciate this important question about our technical starting point. You are correct that Parseval fusion frames provide overcomplete orthonormal bases, but the key distinction lies in _which_ overcomplete bases and _why their specific structure matters_ for parameter updates.
>
> In the first part of section 3 of the paper, we show that LoRA confines the update to a subspace of the parameter space. This motivates a question: _why restrict to a single subspace? Why not use a multitude of them?_ If one were to use multiple subspaces, we would need to
> 1. Train multiple low-rank matrices or
> 2. Derive them from the pretrained matrices (we discuss several drawbacks in response to the question below) or
> 3. Use a predeterministic mechanism to construct these subspaces.
>
> If we decide to proceed with #3, the main question is: how should these subspaces be structured to efficiently capture parameter updates? Fusion frames via Spectral Tetris provide three crucial advantages:
> -  Controlled subspace interactions: The equichordal/equi-isoclinic properties (mentioned in Sec 3.1) ensure that we get optimal geometric separation, prevent redundant representations while allowing good coverage of the parameter space.
> - Algorithmic construction (comes with sparsity): Unlike storing explicit SVD decompositions or learning multiple LoRA-style matrices, fusion frames are generated algorithmically and directly give structured sparsity (see Appendix G and Table 8). This is what gives the high throughput in Table 2. The structure is theoretically elegant, but is very computationally beneficial.
> - Stability: Fusion frame theory gives guarantees that make representations robust to perturbations. This is valuable when we introduce sparsity in the coefficient matrices.
>
> The connection to fine-tuning is direct: we need to decompose parameter updates while maintaining efficiency. Fusion frames provide exactly this structured decomposition, whereas arbitrary overcomplete bases would not give these properties nor the efficiency profile which is one of the main takeaways.
>
> As you acknowledged as one of the strengths of our paper, Fusion Frames are indeed not well studied in the mainstream ML papers. Our work presents the computational advantages of Fusion Frames.

---

> ### Author Response · Authors · 2025-11-21
> **Author reply**
>
> > W.3 The proposed method also looks similar to LoRA-XS (https://arxiv.org/pdf/2405.17604) which fixes the singular vectors and trains the full inner matrix only. But because in theory we can stack multiple modules or multiple heads in LoRA-XS, it can potentially be similar to having a set of overcomplete orthonormal bases in the end?
>
> Thanks for bringing up LoRA-XS because this is a great comparison that can clarify important distinctions. While superficially similar (both use fixed orthonormal bases), both methods differ importantly in how the bases are obtained, what structure they have, and their practical deployment.
>
> We know that LoRA-XS extracts singular vectors from pretrained weights, but this creates scalability barriers:
> 1. Memory. Storing SVD components (left and right singular vector) needs memory. Across a model like Llama-2-7B, this translates to storing more than the base model size (although this can be reduced with engineering effort), and eliminates the main advantage of PEFT methods.
> 2. No sharing. Each layer has unique singular vectors. They must be stored separately. But FrameFT generates dimension-appropriate fusion frames algorithmically ONCE and shares them across all layers of the same dimension. There is zero additional storage.
> 3. Dense versus structured-sparse. We know that SVD produces dense orthonormal matrices. Fusion frames from Spectral Tetris are inherently block-sparse (see Appendix G). There is no post-hoc adjustment, it emerges from the construction. This structural sparsity is why FrameFT achieves 1.7-2x higher throughput than LoRA.
>
> While one could theoretically stack multiple LoRA-XS modules to approximate overcompleteness, this compounds the memory problem above and offers no guarantee of beneficial subspace geometry.
>
> | Method  | **#params** | SST-2    | MRPC     | CoLA     | QNLI     | RTE      | STS-B | Avg      |
> | :-----: | ----------- | -------- | -------- | -------- | -------- | -------- | ----- | -------- |
> | LoRA-XS | 60K         | **96.3** | 91.2     | 68.6     | **94.3** | 89.5     | **92.2**  | **88.7** |
> | FrameFT | 48K         | 96.2     | **92.6** | **69.8** | 93.4     | **88.1** | 91.9  | **88.7** |
>
> FrameFT matches or exceeds LoRA-XS performance with fewer parameters, zero SVD storage overhead, algorithmic generation, and superior inference throughput (not shown above). LoRA-XS adapts to pretrained weight geometry; FrameFT imposes a well-studied geometric structure that is simultaneously efficient by design. This directly enables our efficiency gains.
>
> > W.4 Also, is there any ablation study on how effective the chosen spectral tetris algorithm is? Ie, one can easily think of using randomly generated overcomplete orthonormal basis for Pn and Pm via multiple runs of Gram-Schmidt processes and stacking them. Why spectral tetris?
>
> This is a valid question. We chose Spectral Tetris for one main reason. Spectral Tetris generates fusion frames with inherent block-sparse structure (Appendix G). Random Gram-Schmidt bases are dense. This structural difference (dense versus sparse) directly impacts compute efficiency. In fact, our measured throughput advantage (Table 2) comes from this sparsity during matrix operations. Separately, spectral tetris constructions satisfy equichordal properties, which random bases do not (but this is a minor issue).
>
> In the table below, we present the tokens/sec and the time taken to generate the basis for both methods. The tokens/sec for  Spectral Tetris is much better than a random basis. Additionally, since the Gram-Schmidt process is sequential, it cannot be parallelized on GPUs/TPUs. Hence, the time taken to perform this process is quite high. The standard way to improve this is to use QR decomposition. The basis generation time is a minor issue, as it is amortized across multiple model calls.
>
> |    Model     | Basis              | **tokens/sec** | **Basis generation time (s)** | **QR decomposition time (s)** |
> | :----------: | ------------------ | -------------- | ----------------------------- | ----------------------------- |
> |  Llama-2-7B  | Random Orthonormal | 14.9k          | 112.1                         | 0.31                          |
> |  Llama-2-7B  | Spectral Tetris    | **39.4k**      | **0.09**                      | **0.09**                      |
> |  Gemma-2-9B  | Random Orthonormal | 14.4           | 228.8                         | 0.64                          |
> |  Gemma-2-9B  | Spectral Tetris    | **29.6k**      | **0.21**                      | **0.21**                      |
> | Llama-3.1-8B | Random Orthonormal | 9.5            | 119.6                         | 0.53                          |
> | Llama-3.1-8B | Spectral Tetris    | **27.3k**      | **0.11**                      | **0.11**                      |
>
> In summary, many overcomplete bases may work, but efficient PEFT requires bases that are simultaneously sparse and geometrically well-separated. Spectral Tetris provides both by construction.

---

> > ### Author Response · Authors · 2025-11-21
> > **Author reply**
> >
> > > W.5 Why is Theorem 3.1 (convergence analysis) especially unique for the proposed method? I guess it also holds for LoRA and any other variants (eg, LoRA-XS) if we make some bounded parameter assumptions since most PEFT models are Lipschitz smooth. In this sense, Line 161, where they said LoRA can fail to converge, may be an overstatement?
> >
> > We first answer the easier part of the question and then give the more technical version which we are sure you will appreciate.
> >
> > Our theorem is not automatically applicable to LoRA, and this point is central to our contribution. Why? LoRA has some difficulties with preserving smoothness: Sun (2024b) shows that LoRA's reparameterization breaks Lipschitz smoothness. The authors show that even when the loss $f(W)$ is L-smooth in the weight space, the reparameterized loss $f(A,B)$ is not Lipschitz smooth in the LoRA parameters. Briefly, their result shows that the gradient mapping in LoRA's parameter space can have an unbounded operator norm. FrameFT preserves smoothness: Our Lemma 1 shows that our reparameterization preserves Lipschitz smoothness. When $f(W)$ is L-smooth, we get an explicit bound  = $L \sqrt{r} B_m B_n$. This is not based solely on bounded parameters and is instead a structural property arising from how fusion frames _transforms_ the gradient (see Appendix A, equations 8-13).
> >
> > This matters because now standard convergence theory will apply. For smooth functions, gradient descent with appropriate step size provably converges to critical point. LoRA cannot invoke this result because its reparameterized landscape is non-smooth. Our bound can informed our choices, e.g., choose $B_m = B_n = 1$ to keep the smoothness constant small.
> >
> > Some of the LoRA pathologies are also studied in other works. The asymmetric role of $A$ and $B$ matrices (Malinovsky 2024) comes from their non-smooth interaction. Our symmetric fusion frame structure avoids this. In line 161 we cite (Malinovsky 2024), who proves that LoRA may fail to converge to the optimal solution due to asymmetric parameter dynamics. This is different from converging to some critical point. We will clarify this distinction in revision.
> >
> > In closing, this is not about bounded parameter assumptions, rather a different reparameterization which has different (desirable) smoothness properties.
> >
> > > W.6 Another aspect that is not well discussed in the paper is the memory footprint and the inference efficiency. LoRA's efficient memory use is one of its key benefits: LoRA can only store two low-rank matrices. But the proposed approach needs to store the large overcomplete basis matrices (I guess more than (d x d)) or merge into (d x d) matices, which can be less efficient than LoRA especially for on-device applications where many PEFT methods are targeted for. LoRA can use far less memory footprint during forward pass (run sequentially A@x then B@(A@x)), but the proposed approach uses the overcomplete basis (although fixed during training, but have to be stored as a part of the checkpoints, and loaded at the inference time). The number of trainable parameters may not be a big deal in practice, but the inference time matters more. I guess the proposed method has little benefit in this regard.
> >
> > We should clarify the confusion regarding the storage requirement. Fusion frames are generated algorithmically (O(kd) time via Spectral Tetris) at initialization, not stored in checkpoints. For Llama-2-7B, a LoRA checkpoint requires 67.1 MB (two rank-r matrices per adapted layer), while a FrameFT checkpoint requires only 1.28 MB (sparse coefficients only). This distinction becomes important in multi-task scenarios where one base model serves many task-specific adapters. LoRA requires swapping 67.1 MB per task switch, while FrameFT only needs swapping 1.28 MB per task. Our frames are generated once and reused across all adapters. We hope you agree that this is the practical regime where PEFT methods are deployed at scale.
> >
> > Regarding inference throughput, our measurements in Table 2 show that FrameFT achieves 39.4K tokens/sec on Llama-2-7B compared to LoRA's 23.6K tokens/sec. The throughput advantage comes, as described in our previous answer, from the block-sparse structure (structured sparsity in both $P_m$ and $P_n$). As a comparison, FourierFT with the same parameter count as FrameFT achieves only 1.8K tokens/sec. FrameFT trades a one-time generation cost (amortized across all forward passes) for superior inference speed and much smaller adapter storage.
> >
> >
> > ----
> > Sun (2024) Youbang Sun, Zitao Li, Yaliang Li, and Bolin Ding. Improving loRA in privacy-preserving federated learning, 2024.
> >
> > Malinovsky (2024) Grigory Malinovsky, Umberto Michieli, and others, 2024. Randomized asymmetric chain of LoRA: The first meaningful theoretical framework for low-rank adaptation.

---

> > > ### Author Response · Authors · 2025-11-21
> > > **Author reply**
> > >
> > > > W.7 I am a bit skeptical about the fact that there are too many similar papers these days on sparse PEFT methods that extend LoRA in a variety of different ways to structure the learnable parameter space, eg, Fourier, SVD, vector-based, partially fixing some parameters and varying the rest, etc. Since most ideas are essentially tweaking a bit the parameter space of the (already powerful) pre-trained model, and there is always a high chance that the fine-tuned model performs better than the pre-trained one. So it may be hard to judge which parameter structure is better than others in a uniform manner in a principled way. I mean showing the performance improvements on some selected scenarios (hyperparameter settings and chosen benchmarks) could be potentially possible for most reasonable parametrization. This is not to criticize your paper, but I think some rigorous theoretical study on the relationship between parameter space and model expressibility needs to be done.
> > >
> > > We agree with the concern that there are many PEFT variants. This is why we believe there is a need for a slightly deeper analysis connecting parameterization to empirical expressibility and efficiency. We believe FrameFT offers a principled result beyond empirical tuning on selected benchmarks.
> > >
> > > Our work has three concrete differences to the body of work you mentioned. First, we provide formal convergence analysis showing that FrameFT preserves loss function smoothness, which LoRA lacks. This is a structural guarantee about the optimization landscape that can inform design choices. Second, we target a specific practical scenario where multiple task-specific adapters must be served over a single backbone. The higher throughput compared to LoRA is not a marginal improvement. Third, our construction is algorithmically principled: our frames offer geometric properties and excellent computational structure (block sparsity) by design.
> > >
> > > Our theoretical analysis does generalize to any basis transformation scheme, and the bounds reveal how some frame properties can influence optimization dynamics. This is not a full theory of PEFT, but is nonetheless a concrete step that you were seeking in your comment.
> > >
> > >
> > > We hope that we have addressed all the questions that were raised and provided a better understanding of the benefits of our work and how it differentiates from the current PEFT methods. We are available to answer any further questions you may have.

---

### Meta-Review · Area_Chair_84mN · 2026-01-13

**Summary:**

This paper proposes a new Parameter Efficient Finetuning (PEFT) method called FrameFT. The idea is to represent the weight update with a sparse matrix in a different basis using frames.

As far as I understand, (and the reviewers point out) this idea of using frame theory for PEFT is novel and interesting. Still the paper is not sufficiently polished for publication and I do not think the reviewer's concerns were sufficiently addressed.

**Reviewer Concerns:**

My major concern is the 3rd concern of reviewer TDeP:


There are too many similar sparse PEFT methods that extend LoRA in another space (Fourier, SVD, vector-based, partially fixing some parameters and varying the rest, etc).
To show the benefit over these methods in a rigorous scientific way, one has to show detailed experimental comparisons.

The authors responded that their method has formal convergence guarantees (which is good, even though I was a bit confused about it, see below) but I do not think the experimental validation sufficiently supported the benefit of the frames compared to all these other methods.

**Reviewer Scores:**

Reviewer 6i8s's comments asked questions about the convergence analysis and setup of experiments. I must say that I do not understand the authors argument why their convergence analysis applies globally for non-linear networks, so I am not sure if their response is valid.
If the authors can improve the presentation this can be addressed.

Reviewer TDeP raised the very important concern that despite the elegance of frames, the current peft method sounds very similar to many many others and a more detailed comparison will really help the readers who would like to decide which of the multiple PEFT methods they should use in their problems.

Overall I think this paper has some very interesting ideas and I want to encourage the authors to use these reviews to improve their paper (both theoretically and empirically) and re-submit to the next top ML venue.

---

### Decision · Program_Chairs · 2026-01-26

Reject